# MUVERA: Multi-Vector Retrieval via Fixed Dimensional Encodings

**Laxman Dhulipala**
Google Research and UMD

**Majid Hadian**
Google DeepMind

**Rajesh Jayaram**[*]
Google Research

**Jason Lee**
Google Research

**Vahab Mirrokni**
Google Research

## Abstract

Neural embedding models have become a fundamental component of modern information retrieval (IR) pipelines. These models produce a single embedding $x \in \mathbb{R}^d$ per data-point, allowing for fast retrieval via highly optimized maximum inner product search (MIPS) algorithms. Recently, beginning with the landmark ColBERT paper, *multi-vector models*, which produce a set of embedding per data-point, have achieved markedly superior performance for IR tasks. Unfortunately, using these models for IR is computationally expensive due to the increased complexity of multi-vector retrieval and scoring.

In this paper, we introduce MUVERA (**Mu**lti-**Ve**ctor **R**etrieval **A**lgorithm), a retrieval mechanism which reduces *multi-vector* similarity search to *single-vector* similarity search. This enables the usage of off-the-shelf MIPS solvers for multi-vector retrieval. MUVERA asymmetrically generates *Fixed Dimensional Encodings* (FDEs) of queries and documents, which are vectors whose inner product approximates multi-vector similarity. We prove that FDEs give high-quality $\varepsilon$-approximations, thus providing the first single-vector proxy for multi-vector similarity with theoretical guarantees. Empirically, we find that FDEs achieve the same recall as prior state-of-the-art heuristics while retrieving 2-5× fewer candidates. Compared to prior state of the art implementations, MUVERA achieves consistently good end-to-end recall and latency across a diverse set of the BEIR retrieval datasets, achieving an average of 10% improved recall with 90% lower latency.

## 1 Introduction

Over the past decade, the use of neural embeddings for representing data has become a central tool for information retrieval (IR) [56], among many other tasks such as clustering and classification [39]. Recently, *multi-vector* (MV) representations, introduced by the *late-interaction* framework in ColBERT [29], have been shown to deliver significantly improved performance on popular IR benchmarks. ColBERT and its variants [17, 21, 32, 35, 42, 44, 49, 54] produce *multiple* embeddings per query or document by generating one embedding per token. The query-document similarity is then scored via the *Chamfer Similarity* (§1.1), also known as the MaxSim operation, between the two sets of vectors. These multi-vector representations have many advantages over single-vector (SV) representations, such as better interpretability [15, 50] and generalization [16, 36, 51, 55].

Despite these advantages, multi-vector retrieval is inherently more expensive than single-vector retrieval. Firstly, producing one embedding per token increases the number of embeddings in a dataset by orders of magnitude. Moreover, due to the non-linear Chamfer similarity scoring, there is a lack of optimized systems for multi-vector retrieval. Specifically, single-vector retrieval is

---

[*]Corresponding Author: rkjayaram@google.com

38th Conference on Neural Information Processing Systems (NeurIPS 2024).

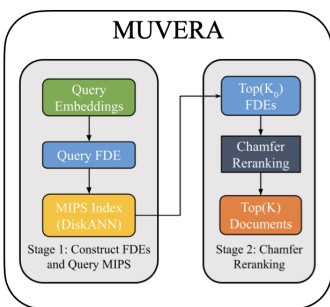
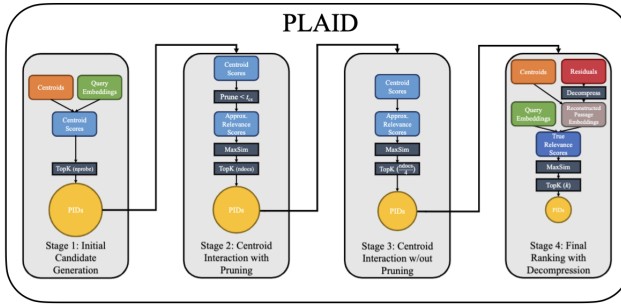

Figure 1: MUVERA's two-step retrieval process, compared to PLAID's multi-stage retrieval process. Diagram on the right from Santhanam et. al. [43] with permission.

generally accomplished via Maximum Inner Product Search (MIPS) algorithms, which have been highly-optimized over the past few decades [18]. However, SV MIPS alone cannot be used for MV retrieval. This is because the MV similarity is the *sum* of the SV similarities of each embedding in a query to the nearest embedding in a document. Thus, a document containing a token with high similarity to a single query token may not be very similar to the query overall. Thus, in an effort to close the gap between SV and MV retrieval, there has been considerable work in recent years to design custom MV retrieval algorithms with improved efficiency [12, 21, 42, 43].

The most prominent approach to MV retrieval is to employ a multi-stage pipeline beginning with single-vector MIPS. The basic version of this approach is as follows: in the initial stage, the most similar document tokens are found for each of the query tokens using SV MIPS. Then the corresponding documents containing these tokens are gathered together and rescored with the original Chamfer similarity. We refer to this method as the *single-vector heuristic*. ColBERTv2 [44] and its optimized retrieval engine PLAID [43] are based on this approach, with the addition of several intermediate stages of pruning. In particular, PLAID employs a complex *four*-stage retrieval and pruning process to gradually reduce the number of final candidates to be scored (Figure 1). Unfortunately, as described above, employing SV MIPS on individual query embeddings can fail to find the true MV nearest neighbors. Additionally, this process is expensive, since it requires querying a significantly larger MIPS index for *every* query embedding (larger because there are multiple embeddings per document). Finally, these multi-stage pipelines are complex and highly sensitive to parameter setting, as recently demonstrated in a reproducibility study [37], making them difficult to tune. To address these challenges and bridge the gap between single and multi-vector retrieval, in this paper we seek to design faster and simplified MV retrieval algorithms.

**Contributions.** We propose MUVERA: a multi-vector retrieval mechanism based on a light-weight and provably correct reduction to single-vector MIPS. MUVERA employs a fast, data-oblivious transformation from a set of vectors to a single vector, allowing for retrieval via highly-optimized MIPS solvers before a single stage of re-ranking. Specifically, MUVERA transforms query and document MV sets $Q, P \subset \mathbb{R}^d$ into single fixed-dimensional vectors $\vec{q}, \vec{p}$, called *Fixed Dimensional Encodings* (FDEs), such that the the dot product $\vec{q} \cdot \vec{p}$ approximates the multi-vector similarity between $Q, P$ (§2). Empirically, we show that retrieving with respect to the FDE dot product significantly outperforms the single vector heuristic at recovering the Chamfer nearest neighbors (§3.1). For instance, on MS MARCO, our FDEs Recall@$N$ surpasses the Recall@2-5N achieved by the SV heuristic while scanning a similar total number of floats in the search.

We prove in (§2.1) that our FDEs have strong approximation guarantees; specifically, the FDE dot product gives an $\varepsilon$-approximation to the true MV similarity. This gives the first algorithm with provable guarantees for Chamfer similarity search with strictly faster than brute-force runtime (Theorem 2.2). Thus, MUVERA provides the first principled method for MV retrieval via a SV proxy.

We compare the end-to-end retrieval performance of MUVERA to PLAID on several of the BEIR IR datasets, including the well-studied MS MARCO dataset. We find MUVERA to be a robust and efficient retrieval mechanism; across the datasets we evaluated, MUVERA obtains an average of 10% higher recall, while requiring 90% lower latency on average compared with PLAID. Additionally, MUVERA crucially incorporates a vector compression technique called *product quantization* that enables us to compress the FDEs by 32× (i.e., storing 10240 dimensional FDEs using 1280 bytes) while incurring negligible quality loss, resulting in a significantly smaller memory footprint.

## 1.1 Chamfer Similarity and the Multi-Vector Retrieval Problem

Given two sets of vectors $Q, P \subset \mathbb{R}^d$, the *Chamfer Similarity* is given by

$$\text{CHAMFER}(Q, P) = \sum_{q \in Q} \max_{p \in P} \langle q, p \rangle$$

where $\langle \cdot, \cdot \rangle$ is the standard vector inner product. Chamfer similarity is the default method of MV similarity used in the *late-interaction* architecture of ColBERT, which includes systems like ColBERTv2 [44], Baleen [28], Hindsight [41], DrDecr [34], and XTR [32], among many others. These models encode queries and documents as sets $Q, P \subset \mathbb{R}^d$ (respectively), where the query-document similarity is given by $\text{CHAMFER}(Q, P)$. We note that Chamfer Similarity (and its distance variant) itself has a long history of study in the computer vision (e.g., [4, 6, 14, 27, 45]) and graphics [33] communities, and had been previously used in the ML literature to compare sets of embeddings [3, 5, 30, 48]. In these works, Chamfer is also referred to as *MaxSim* or the *relaxed earth mover distance*; we choose the terminology *Chamfer* due to its historical precedence [6].

In this paper, we study the problem of Nearest Neighbor Search (NNS) with respect to the Chamfer Similarity. Specifically, we are given a dataset $D = \{P_1, \ldots, P_n\}$ where each $P_i \subset \mathbb{R}^d$ is a set of vectors. Given a query subset $Q \subset \mathbb{R}^d$, the goal is to quickly recover the nearest neighbor $P^* \in D$, namely:

$$P^* = \arg \max_{P_i \in D} \text{CHAMFER}(Q, P_i)$$

For the retrieval system to be scalable, this must be achieved in time significantly faster than brute-force scoring each of the $n$ similarities $\text{CHAMFER}(Q, P_i)$ within $D$.

## 1.2 Our Approach: Reducing Multi-Vector Search to Single-Vector MIPS

MUVERA is a streamlined procedure that directly reduces the Chamfer Similarity Search to MIPS. For a pre-specified target dimension $d_{\text{FDE}}$, MUVERA produces randomized mappings $\mathbf{F}_q : 2^{\mathbb{R}^d} \to \mathbb{R}^{d_{\text{FDE}}}$ (for queries) and $\mathbf{F}_{\text{doc}} : 2^{\mathbb{R}^d} \to \mathbb{R}^{d_{\text{FDE}}}$ (for documents) such that, for all query and document multi-vector representations $Q, P \subset \mathbb{R}^d$, we have:

$$\langle \mathbf{F}_q(Q), \mathbf{F}_{\text{doc}}(P) \rangle \approx \text{CHAMFER}(Q, P)$$

We refer to the vectors $\mathbf{F}_q(Q), \mathbf{F}_{\text{doc}}(P)$ as *Fixed Dimensional Encodings* (FDEs). MUVERA first applies $\mathbf{F}_{\text{doc}}$ to each document representation $P \in D$, and indexes the set $\{\mathbf{F}_{\text{doc}}(P)\}_{P \in D}$ into a MIPS solver. Given a query $Q \subset \mathbb{R}^d$, MUVERA quickly computes $\mathbf{F}_q(Q)$ and feeds it to the MIPS solver to recover the top-$k$ most similar document FDE's $\mathbf{F}_{\text{doc}}(P)$. Finally, we re-rank these candidates by the original Chamfer similarity. See Figure 1 for an overview. We remark that one important advantage of the FDEs is that the functions $\mathbf{F}_q, \mathbf{F}_{\text{doc}}$ are *data-oblivious*, making them robust to distribution shifts, and easily usable in streaming settings.

## 1.3 Related Work on Multi-Vector Retrieval

The early multi-vector retrieval systems, such as ColBERT [29], all implement optimizations of the previously described SV heuristic, where the initial set of candidates is found by querying a MIPS index for every query token $q \in Q$. In ColBERTv2 [44], the document token embeddings are first clustered via k-means, and the first round of scoring uses cluster centroids instead of the original token. This technique was further optimized in PLAID [43] by employing a four-stage pipeline to progressively prune candidates before a final re-ranking (Figure 1).

An alternative approach with proposed in DESSERT [12], whose authors also pointed out the limitations of the SV heuristic, and proposed an algorithm based on Locality Sensitive Hashing (LSH) [20]. They prove that their algorithm recovers $\varepsilon$-approximate nearest neighbors in time $\tilde{O}(n|Q|T)$, where $T$ is roughly the maximum number of document tokens $p \in P_i$ that are similar to any query token $q \in Q$, which can be as large as $\max_i |P_i|$. Thus, in the worst case, their algorithm runs no faster than brute-force. Conversely, our algorithm recovers $\varepsilon$-approximate nearest neighbors and *always* runs in time $\tilde{O}(n|Q|)$. Experimentally, DESSERT is 2-5× faster than PLAID, but attains worse recall (e.g. 2-2.5% R@1000 on MS MARCO). Conversely, we match and sometimes strongly exceed PLAID's recall with up to 5.7× lower latency. Additionally, DESSERT still employs an initial filtering stage based on $k$-means clustering of individual query token embeddings (in the manner of ColBERTv2), thus they do not truly avoid the aforementioned limitations of the SV heuristic.

## 2 Fixed Dimensional Encodings

We now describe our process for generating FDEs. Our transformation is reminiscent of the technique of probabilistic tree embeddings [1, 7, 10, 13], which can be used to transform a set of vectors into a single vector. For instance, they have been used to embed the Earth Mover's Distance into the $\ell_1$ metric [1, 10, 22, 24], and to embed the weight of a Euclidean MST of a set of vectors into the Hamming metric [9, 22, 23]. However, since we are working with inner products, which are not metrics, instead of $\ell_p$ distances, an alternative approach for our transformation will be needed.

The intuition behind our transformation is as follows. Hypothetically, for two MV representations $Q, P \subset \mathbb{R}^d$, if we knew the optimal mapping $\pi : Q \to P$ in which to match them, then we could create vectors $\vec{q}, \vec{p}$ by concatenating all the vectors in $Q$ and their corresponding images in $P$ together, so that $\langle \vec{q}, \vec{p} \rangle = \sum_{q \in Q} \langle q, \pi(q) \rangle = \text{CHAMFER}(Q, P)$. However, since we do not know $\pi$ in advance, and since different query-document pairs have different optimal mappings, this simple concatenation clearly will not work. Instead, our goal is to find a randomized ordering over *all* the points in $\mathbb{R}^d$ so that, after clustering close points together, the dot product of *any* query-document pair $Q, P \subset \mathbb{R}^d$ concatenated into a single vector under this ordering will approximate the Chamfer similarity.

The first step is to partition the latent space $\mathbb{R}^d$ into $B$ clusters so that vectors that are closer are more likely to land in the same cluster. Let $\varphi : \mathbb{R}^d \to [B]$ be such a partition (for an integer $N \geqslant 1$ we use the notation $[N] = \{1, 2, \ldots, N\}$); $\varphi$ can be implemented via Locality Sensitive Hashing (LSH) [20], $k$-means, or other methods; we discuss choices for $\varphi$ later in this section. After partitioning via $\varphi$, the hope is that for each $q \in Q$, the closest $p \in P$ lands in the same cluster (i.e. $\varphi(q) = \varphi(p)$). Hypothetically, if this were to occur, then:

$$\text{CHAMFER}(Q, P) = \sum_{k=1}^{B} \sum_{\substack{q \in Q \\ \varphi(q)=k}} \max_{\substack{p \in P \\ \varphi(p)=k}} \langle q, p \rangle \tag{1}$$

If $p$ is the only point in $P$ that collides with $q$, then (1) can be realized as a dot product between two vectors $\vec{q}, \vec{p}$ by creating one block of $d$ coordinates in $\vec{q}, \vec{p}$ for each cluster $k \in [B]$ (call these blocks $\vec{q}_{(k)}, \vec{p}_{(k)} \in \mathbb{R}^d$), and setting $\vec{q}_{(k)}, \vec{p}_{(k)}$ to be the sum of all $q \in Q$ (resp. $p \in P$) that land in the $k$-th cluster under $\varphi$. However, if multiple $p' \in P$ collide with $q$, then $\langle \vec{q}, \vec{p} \rangle$ will differ from (1), since *every* $p'$ with $\varphi(p') = \varphi(q)$ will contribute at least $\langle q, p' \rangle$ to $\langle \vec{q}, \vec{p} \rangle$. To resolve this, we set $\vec{p}_{(k)}$ to be the *centroid* of the $p \in P$'s with $\varphi(p) = \varphi(q)$. Formally, for $k = 1, \ldots, B$, we define

$$\vec{q}_{(k)} = \sum_{\substack{q \in Q \\ \varphi(q)=k}} q, \qquad \vec{p}_{(k)} = \frac{1}{|P \cap \varphi^{-1}(k)|} \sum_{\substack{p \in P \\ \varphi(p)=k}} p \tag{2}$$

Setting $\vec{q} = (\vec{q}_{(1)}, \ldots, \vec{q}_{(B)})$ and $\vec{p} = (\vec{p}_{(1)}, \ldots, \vec{p}_{(B)})$, then we have

$$\langle \vec{q}, \vec{p} \rangle = \sum_{k=1}^{B} \sum_{\substack{q \in Q \\ \varphi(q)=k}} \frac{1}{|P \cap \varphi^{-1}(k)|} \sum_{\substack{p \in P \\ \varphi(p)=k}} \langle q, p \rangle \tag{3}$$

Note that the resulting dimension of the vectors $\vec{q}, \vec{p}$ is $dB$. To reduce the dependency on $d$, we can apply a random linear projection $\psi : \mathbb{R}^d \to \mathbb{R}^{d_{\text{proj}}}$ to each block $\vec{q}_{(k)}, \vec{p}_{(k)}$, where $d_{\text{proj}} < d$. Specifically, we define $\psi(x) = (1/\sqrt{d_{\text{proj}}})Sx$, where $S \in \mathbb{R}^{d_{\text{proj}} \times d}$ is a random matrix with uniformly distributed $\pm 1$ entries. We can then define $\vec{q}_{(k),\psi} = \psi(\vec{q}_{(k)})$ and $\vec{p}_{(k),\psi} = \psi(\vec{p}_{(k)})$, and define the *FDE's with inner projection* as $\vec{q}_\psi = (\vec{q}_{(1),\psi}, \ldots, \vec{q}_{(B),\psi})$ and $\vec{p}_\psi = (\vec{p}_{(1),\psi}, \ldots, \vec{p}_{(B),\psi})$. When $d = d_{\text{proj}}$, we simply define $\psi$ to be the identity mapping, in which case $\vec{q}_\psi, \vec{p}_\psi$ are identical to $\vec{q}, \vec{p}$. To increase accuracy of (3) in approximating (1), we repeat the above process $R_{\text{reps}} \geqslant 1$ times independently, using different randomized partitions $\varphi_1, \ldots, \varphi_{R_{\text{reps}}}$ and projections $\psi_1, \ldots, \psi_{R_{\text{reps}}}$. We denote the vectors resulting from $i$-th repetition by $\vec{q}_{i,\psi}, \vec{p}_{i,\psi}$. Finally, we concatenate these $R_{\text{reps}}$ vectors together, so that our final FDEs are defined as $\mathbf{F}_{\text{q}}(Q) = (\vec{q}_{1,\psi}, \ldots, \vec{q}_{R_{\text{reps}},\psi})$ and $\mathbf{F}_{\text{doc}}(P) = (\vec{p}_{1,\psi}, \ldots, \vec{p}_{R_{\text{reps}},\psi})$. Observe that a complete FDE mapping is specified by the three parameters $(B, d_{\text{proj}}, R_{\text{reps}})$, resulting in a final dimension of $d_{\text{FDE}} = B \cdot d_{\text{proj}} \cdot R_{\text{reps}}$.

**Choice of Space Partition.** When choosing the partition function $\varphi$, the desired property is that points are more likely to collide (i.e. $\varphi(x) = \varphi(y)$) the closer they are to each other. Such functions

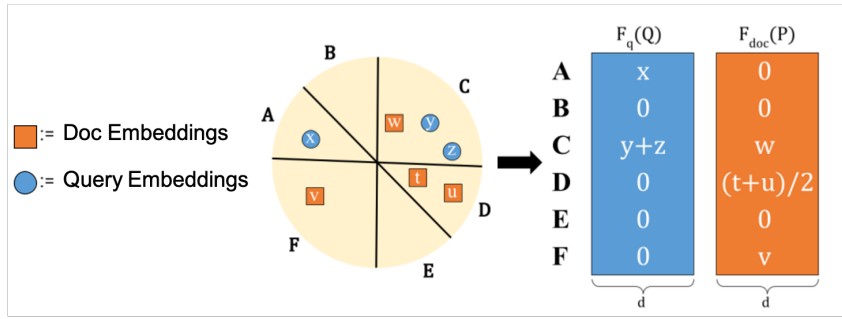

Figure 2: FDE Generation Process. Three SimHashes ($k_{\text{sim}} = 3$) split space into six regions labelled $A$-$F$ (in high-dimensions $B = 2^{k_{\text{sim}}}$, but $B = 6$ here since $d = 2$). $\mathbf{F}_q(Q), \mathbf{F}_{\text{doc}}(P)$ are shown as $B \times d$ matrices, where the $k$-th row is $\vec{q}_{(k)}, \vec{p}_{(k)}$. The actual FDEs are flattened versions of these matrices. Not shown: inner projections, repetitions, and `fill_empty_clusters`.

with this property exist, and are known as *locality-sensitive hash functions* (LSH) (see [20]). When the vectors are normalized, as they are for those produced by ColBERT-style models, SimHash [8] is the standard choice of LSH. Specifically, for any $k_{\text{sim}} \geqslant 1$, we sample random Gaussian vectors $g_1, \ldots, g_{k_{\text{sim}}} \in \mathbb{R}^d$, and set $\boldsymbol{\varphi}(x) = (\mathbf{1}(\langle g_1, x \rangle > 0), \ldots, \mathbf{1}(\langle g_{k_{\text{sim}}}, x \rangle > 0))$, where $\mathbf{1}(\cdot) \in \{0, 1\}$ is the indicator function. Converting the bit-string to decimal, $\boldsymbol{\varphi}(x)$ gives a mapping from $\mathbb{R}^d$ to $[B]$ where $B = 2^{k_{\text{sim}}}$. In other words, SimHash partitions $\mathbb{R}^d$ by drawing $k_{\text{sim}}$ random half-spaces, and each of the $2^{k_{\text{sim}}}$ clusters is formed by the $k_{\text{sim}}$-wise intersection of each of these halfspaces or their complement. Another natural approach is to choose $k_{\text{CENTER}} \geqslant 1$ centers from the collection of all token embeddings $\cup_{i=1}^n P_i$, either randomly or via $k$-means, and set $\boldsymbol{\varphi}(x) \in [k_{\text{CENTER}}]$ to be the index of the center nearest to $x$. We compare this method to SimHash in (§3.1).

**Filling Empty Clusters.** A key source of error in the FDE's approximation is when the nearest vector $p \in P$ to a given query embedding $q \in Q$ maps to a different cluster, namely $\boldsymbol{\varphi}(p) \neq \boldsymbol{\varphi}(q) = k$. This can be made less likely by decreasing $B$, at the cost of making it more likely for other $p' \in P$ to also map to the same cluster, moving the centroid $\vec{p}_{(k)}$ farther from $p$. If we increase $B$ too much, it is possible that no $p \in P$ collides with $\boldsymbol{\varphi}(q)$. To avoid this trade-off, we directly ensure that if no $p \in P$ maps to a cluster $k$, then instead of setting $\vec{p}_{(k)} = 0$ we set $\vec{p}_{(k)}$ to the point $p$ that is *closest* to cluster $k$. As a result, increasing $B$ will result in a more accurate estimator, as this results in smaller clusters. Formally, for any cluster $k$ with $P \cap \boldsymbol{\varphi}^{-1}(k) = \emptyset$, if `fill_empty_clusters` is enabled, we set $\vec{p}_{(k)} = p$ where $p \in P$ is the point for which $\boldsymbol{\varphi}(p)$ has the fewest number of disagreeing bits with $k$ (both thought of as binary strings), with ties broken arbitrarily. We do not enable this for query FDEs, as doing so would result in a given $q \in Q$ contributing to the dot product multiple times.

**Final Projections.** A natural approach to reducing the dimensionality is to apply a final projection $\boldsymbol{\psi}' : \mathbb{R}^{d_{\text{FDE}}} \to \mathbb{R}^{d_{\text{final}}}$ (also implemented via multiplication by a random $\pm 1$ matrix) to the FDE's, reducing the final dimensionality to any $d_{\text{final}} < d_{\text{FDE}}$. Experimentally, we find that final projections can provide small but non-trivial 1-2% recall boosts for a fixed dimension (see §C.2).

## 2.1 Theoretical Guarantees for FDEs

We now state our theoretical guarantees for our FDE construction. For clarity, we state our results in terms of normalized Chamfer similarity $\text{NCHAMFER}(Q, P) = \frac{1}{|Q|}\text{CHAMFER}(Q, P)$. This ensures $\text{NCHAMFER}(Q, P) \in [-1, 1]$ whenever the vectors in $Q, P$ are normalized. Note that this factor of $1/|Q|$ does not affect the relative scoring of documents for a fixed query. In what follows, we assume that all token embeddings are normalized (i.e. $\|q\|_2 = \|p\|_2 = 1$ for all $q \in Q, p \in P$). Note that ColBERT-style late interaction MV models indeed produce normalized token embeddings. We will always use the `fill_empty_clusters` method for document FDEs, but never for queries.

Our main result is that FDEs give $\varepsilon$-additive approximations of the Chamfer similarity. The proof uses the properties of LSH (SimHash) to show that for each query point $q \in Q$, the point $q$ gets mapped to a cluster $\varphi(q)$ that *only* contains points $p \in P$ that are close to $q$ (within $\varepsilon$ of the closest point to $q$); the fact that at least one point collides with $q$ uses the `fill_empty_partitions` method.

**Theorem 2.1** (FDE Approximation). *Fix any $\varepsilon, \delta > 0$, and sets $Q, P \subset \mathbb{R}^d$ of unit vectors, and let $m = |Q| + |P|$. Then setting $k_{\text{sim}} = O\left(\frac{\log(m\delta^{-1})}{\varepsilon}\right)$, $d_{\text{proj}} = O\left(\frac{1}{\varepsilon^2}\log(\frac{m}{\varepsilon\delta})\right)$, $R_{\text{reps}} = 1$, so that*

$d_{FDE} = (m/\delta)^{O(1/\varepsilon)}$, then in expectation and with probability at least $1 - \delta$ we have

$$\text{NChamfer}(Q, P) - \varepsilon \leqslant \frac{1}{|Q|} \langle \mathbf{F}_q(Q), \mathbf{F}_{doc}(P) \rangle \leqslant \text{NChamfer}(Q, P) + \varepsilon$$

Finally, we show that our FDE's give an $\varepsilon$-approximate solution to Chamfer similarity search, using FDE dimension that depends only *logarithmically* on the size of the dataset $n$. Using the fact that our query FDEs are sparse (Lemma A.1), one can run exact MIPS over the FDEs in time $\tilde{O}(|Q| \cdot n)$, improving on the brute-force runtime of $O(|Q| \max_i |P_i| n)$ for Chamfer similarity search.

**Theorem 2.2.** *Fix any $\varepsilon > 0$, query $Q$, and dataset $P = \{P_1, \ldots, P_n\}$, where $Q \subset \mathbb{R}^d$ and each $P_i \subset \mathbb{R}^d$ is a set of unit vectors. Let $m = |Q| + \max_{i \in [n]} |P_i|$. Let $k_{sim} = O(\frac{\log m}{\varepsilon})$, $d_{proj} = O(\frac{1}{\varepsilon^2} \log(m/\varepsilon))$ and $R_{reps} = O(\frac{1}{\varepsilon^2} \log n)$ so that $d_{FDE} = m^{O(1/\varepsilon)} \cdot \log n$. Then if $i^* = \arg\max_{i \in [n]} \langle \mathbf{F}_q(Q), \mathbf{F}_{doc}(P_i) \rangle$, with high probability (i.e. $1 - 1/\text{poly}(n)$) we have:*

$$\text{NChamfer}(Q, P_{i^*}) \geqslant \max_{i \in [n]} \text{NChamfer}(Q, P_i) - \varepsilon$$

*Given the query $Q$, the document $P^*$ can be recovered in time $O\left(|Q| \max\{d, n\} \frac{1}{\varepsilon^4} \log(\frac{m}{\varepsilon}) \log n\right)$.*

## 3 Evaluation

In this section, we evaluate our FDEs as a method for MV retrieval. First, we evaluate the FDEs themselves (offline) as a proxy for Chamfer similarity (§3.1). In (§3.2), we discuss the implementation of MUVERA, as well as several optimizations made in the search. Then we evaluate the latency of MUVERA compared to PLAID, and study the effects of the aforementioned optimizations.

**Datasets.** Our evaluation includes results from six of the well-studied BEIR [46] information retrieval datasets: MS MARCO [40] (CC BY-SA 4.0), HotpotQA (CC BY-SA 4.0) [53], NQ (Apache-2.0) [31], Quora (Apache-2.0) [46], SciDocs (CC BY 4.0) [11], and ArguAna (Apache-2.0) [47]. These datasets were selected for varying corpus size (8K-8.8M) and average number of document tokens (18-165); see (§B) for further dataset statistics. Following [43], we use the development set for our experiments on MS MARCO, and use the test set on the other datasets.

**MV Model, MV Embedding Sizes, and FDE Dimensionality.** We compute our FDEs on the MV embeddings produced by the ColBERTv2 model [44] (MIT License), which have a dimension of $d = 128$ and a fixed number $|Q| = 32$ of embeddings per query. The number of document embeddings is variable, ranging from an average of 18.3 on Quora to 165 on Scidocs. This results in 2,300-21,000 floats per document on average (e.g. 10,087 for MS MARCO). Thus, when constructing our FDEs we consider a comparable range of dimensions $d_{\text{FDE}}$ between 1,000-20,000. Furthermore, using product quantization, we show in (§3.2) that the FDEs can be significantly compressed by $32\times$ with minimal quality loss, further increasing the practicality of FDEs.

### 3.1 Offline Evaluation of FDE Quality

We evaluate the quality of our FDEs as a proxy for the Chamfer similarity, without any re-ranking and using exact (offline) search. We first demonstrate that FDE recall quality improves dependably as the dimension $d_{\text{FDE}}$ increases, making our method relatively easy to tune. We then show that FDEs are a more effective method of retrieval than the SV heuristic. Specifically, the FDE method achieves Recall@$N$ exceeding the Recall@2-4N of the SV heuristic, while in principle scanning a similar number of floats in the search. This suggests that the success of the SV heuristic is largely due to the significant effort put towards optimizing it (as supported by [37]), and similar effort for FDEs may result in even bigger efficiency gains. Additional plots can be found in (§C). All recall curves use a single FDE instantiation, since in (§C.1) we show the variance of FDE recall is negligible.

**FDE Quality vs. Dimensionality.** We study how the retrieval quality of FDE's improves as a function of the dimension $d_{\text{FDE}}$. We perform a grid search over FDE parameters $R_{\text{reps}} \in \{1, 5, 10, 15, 20\}, k_{\text{sim}} \in \{2, 3, 4, 5, 6\}, d_{\text{proj}} \in \{8, 16, 32, 64\}$, and compute recall on MS MARCO (Figure 3). We find that Pareto optimal parameters are generally achieved by larger $R_{\text{reps}}$, with $k_{\text{sim}}, d_{\text{proj}}$ playing a lesser role in improving quality. Specifically, $(R_{\text{reps}}, k_{\text{sim}}, d_{\text{proj}}) \in \{(20, 3, 8), (20, 4, 8)(20, 5, 8), (20, 5, 16)\}$ were all Pareto optimal for their respective dimensions (namely $R_{\text{reps}} \cdot 2^{k_{\text{sim}}} \cdot d_{\text{proj}}$). While there are small variations depending on the parameter choice, the FDE quality is tightly linked to dimensionality; increase in dimensionality will generally result in

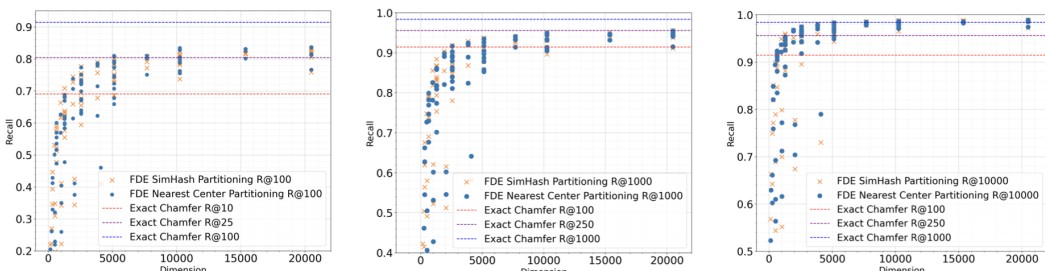

Figure 3: FDE recall vs dimension for varying FDE parameters on MS MARCO. Plots show FDE Recall@100,1k,10k left to right. Recalls@$N$ for exact Chamfer scoring is shown by dotted lines.

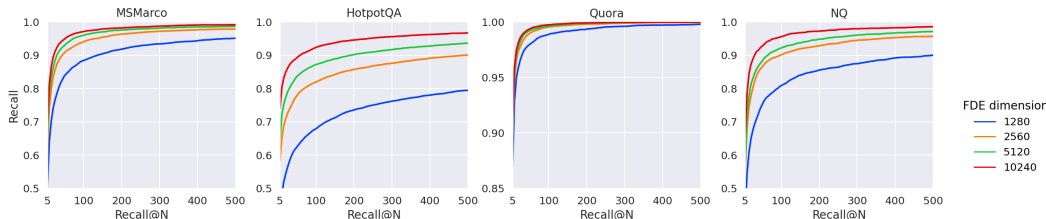

Figure 4: Comparison of FDE recall versus brute-force search over Chamfer similarity.

quality gains. We also evaluate using $k$-means as a method of partitioning instead of SimHash. Specifically, we cluster the document embeddings with $k$-means and set $\varphi(x)$ to be the index of the nearest centroid to $x$. We perform a grid search over the same parameters (but with $k \in \{4, 8, 16, 32, 64\}$ to match $B = 2^{k_{\text{sim}}}$). We find that $k$-means partitioning offers no quality gains on the Pareto Frontier over SimHash, and is often worse. Moreover, FDE construction with $k$-means is no longer data oblivious. Thus, SimHash is chosen as the preferred method for partitioning for the remainder of our experiments.

In Figure 4, we evaluate the FDE retrieval quality with respect to the Chamfer similarity (instead of labelled ground truth data). We compute 1Recall@$N$, which is the fraction of queries for which the Chamfer 1-nearest neighbor is among the top-$N$ most similar in FDE dot product. We choose FDE parameters which are Pareto optimal for the dimension from the above grid search. We find that FDE's with fewer dimensions that the original MV representations achieve significantly good recall across multiple BEIR retrieval datasets. For instance, on MS MARCO (where $d \cdot m_{avg} \approx 10K$) we achieve 95% recall while retrieving only 75 candidates using $d_{\text{FDE}} = 5120$.

**Single Vector Heuristic vs. FDE retrieval.** We compare the quality of FDEs as a proxy for retrieval against the previously described SV heuristic, which is the method underpinning PLAID. Recall that in this method, for each of the $i = 1, \ldots, 32$ query vectors $q_i$ we compute the $k$ nearest neighbors $p_{1,i}, \ldots, p_{k,i}$ from the set $\cup_i P_i$ of all document token embeddings. To compute Recall@$N$, we create an ordered list $\ell_{1,1}, \ldots, \ell_{1,32}, \ell_{2,1}, \ldots$, where $\ell_{i,j}$ is the document ID containing $p_{i,j}$, consisting of the 1-nearest neighbors of the queries, then the 2-nearest neighbors, and so on. When re-ranking, one first removes duplicate document IDs from this list. Since duplicates cannot be detected while performing the initial 32 SV MIPS queries, the SV heuristic needs to over-retrieve to reach a desired number of unique candidates. Thus, we note that the true recall curve of implementations of the SV heuristic (e.g. PLAID) is somewhere between the case of no deduplication and full deduplication; we compare to both in Figure 5.

To compare the cost of the SV heuristic to running MIPS over the FDEs, we consider the total number of floats scanned by both using a brute force search. The FDE method must scan $n \cdot d_{\text{FDE}}$ floats to compute the $k$-nearest neighbors. For the SV heuristic, one runs 32 brute force scans over $n \cdot m_{avg}$ vectors in 128 dimensions, where $m_{avg}$ is the average number of embeddings per document (see §B for values of $m_{avg}$). For MS MARCO, where $m_{avg} = 78.8$, the SV heuristic searches through $32 \cdot 128 \cdot 78.8 \cdot n$ floats. This allows for an FDE dimension of $d_{\text{FDE}} = 322,764$ to have comparable cost! We can extend this comparison to fast approximate search – suppose that approximate MIPS

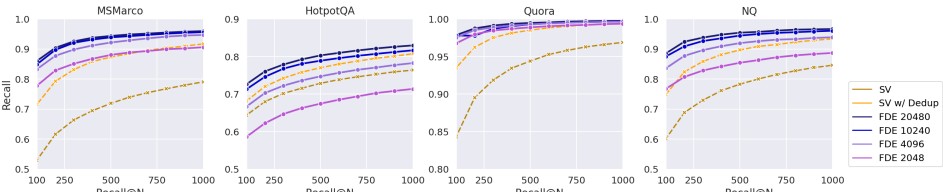

Figure 5: FDE retrieval vs SV Heuristic, both with and without document id deduplication.

over $n$ vectors can be accomplished in sublinear $n^\varepsilon$ time, for some $\varepsilon \in (0,1)$. Then even in the unrealistic case of $\varepsilon = 0$, we can still afford an FDE dimension of $d_{\text{FDE}} = 32 \cdot 128 = 4096$.

The results can be found in Figure 5. We build FDEs once for each dimension, using $R_{\text{reps}} = 40, k_{\text{sim}} = 6, d_{\text{proj}} = d = 128$, and then applying a final projection to reduce to the target dimension (see C.2 for experiments on the impact of final projections). On MS MARCO, even the 4096-dimensional FDEs match the recall of the (deduplicated) SV heuristic while retrieving 1.75-3.75× *fewer* candidates (our Recall@$N$ matches the Recall@1.75-3.75$N$ of the SV heuristic), and 10.5-15× fewer than to the non-deduplicated SV heuristic. For our 10240-dimension FDEs, these numbers are 2.6-5× and 20-22.5× fewer, respectively. For instance, we achieve $80\%$ recall with 60 candidates when $d_{\text{FDE}} = 10240$ and 80 candidates when $d_{\text{FDE}} = 4096$, but the SV heuristic requires 300 and 1200 candidates (for dedup and non-dedup respectively). See Table 1 for further comparisons.

**Variance.** Note that although the FDE generation is a randomized process, we show in (§C.1) that the variance of the FDE Recall is essentially negligible; for instance, the standard deviation Recall@1000 is at most $0.08$-$0.16\%$ for FDEs with 2-10$k$ dimensions.

### 3.2 Online Implementation and End-to-End Evaluation

We implemented MUVERA, an FDE generation and end-to-end retrieval engine in C++. We discussed FDE generation and various optimizations and their tradeoffs (§3.1). Next, we discuss how we perform retrieval over the FDEs, and additional optimizations.

**Single-Vector MIPS Retrieval using DiskANN.** Our single-vector retrieval engine uses a scalable implementation [38] of DiskANN [25] (MIT License), a state-of-the-art graph-based ANNS algorithm. We build DiskANN indices by using the uncompressed document FDEs with a maximum degree of 200 and a build beam-width of 600. Our retrieval works by querying the DiskANN index using beam search with beam-width $W$, and subsequently reranking the retrieved candidates with Chamfer similarity. The only tuning knob in our system is $W$; increasing $W$ increases the number of candidates retrieved by MUVERA, which improves the recall.

**Product Quantization (PQ).** To further improve the memory usage of MUVERA, we use a textbook vector compression technique called product quantization (PQ) with asymmetric querying [19, 26] on the FDEs. We refer to product quantization with $C$ centers per group of $G$ dimensions as PQ-$C$-$G$. For example, PQ-256-8, which we find to provide the best tradeoff between quality and compression in our experiments, compresses every consecutive set of 8 dimensions to one of 256 centers. Thus PQ-256-8 provides 32× compression over storing each dimension using a single float, since each block of 8 floats is represented by a single byte. See (§C.4) for further experiments and details on PQ.

**Experimental Setup.** We run our online experiments on an Intel Sapphire Rapids machine on Google Cloud (c3-standard-176). The machine supports up to 176 hyper-threads. We run latency experiments using a single thread, and run our QPS experiments on all 176 threads.

**Ball Carving.** To improve re-ranking speed, we reduce the number of query embeddings by clustering them via a *ball carving* method and replacing the embeddings in each cluster with their sum. This speeds up reranking without decreasing recall. Specifically, we group the queries $Q$ into clusters $C_1, \ldots, C_k$, setting $c_i = \sum_{q \in C_i} q$ and $Q_C = \{c_1, \ldots, c_k\}$. Then, after retrieving a set of candidate documents with the FDEs, instead of rescoring via $\text{CHAMFER}(Q, P)$ for each candidate $P$, we rescore via $\text{CHAMFER}(Q_C, P)$, which runs in time $O(|Q_C| \cdot |P|)$, offering speed-ups when the number of clusters is small. Instead of fixing $k$, we perform greedy ball-carving to allow $k$ to adapt to $Q$. Specifically, given a threshold $\tau$, we select an arbitrary point $q \in Q$, cluster it with all other points $q' \in Q$ with $\langle q, q' \rangle \geqslant \tau$, remove the clustered points and repeat until all points are clustered.

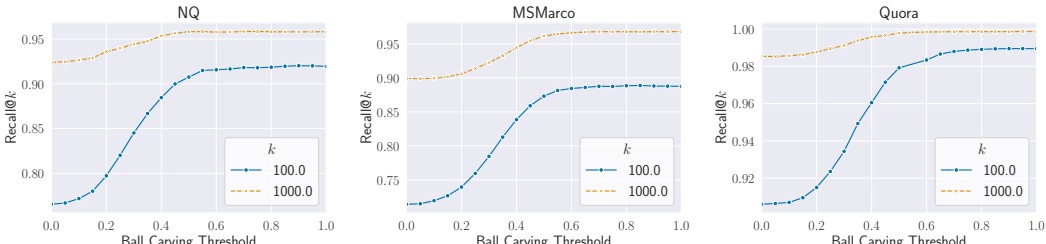

Figure 6: Plots showing the trade-off between the threshold used for ball carving and the end-to-end recall.

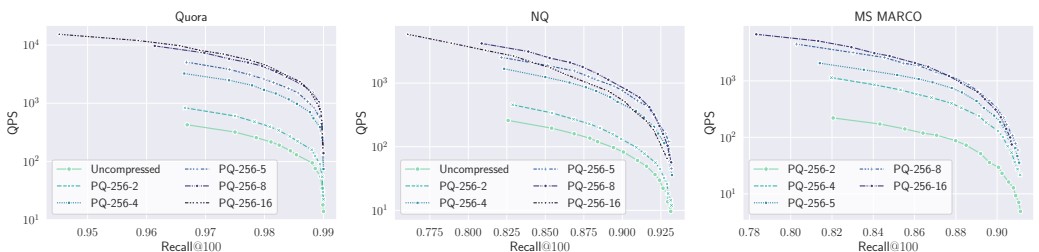

Figure 7: Plots showing the QPS vs. Recall@100 for MUVERA on a subset of the BEIR datasets. The different curves are obtained by using different PQ methods on 10240-dimensional FDEs.

In Figure 6, we show the the trade-off between end-to-end Recall@$k$ of MUVERA and the ball carving threshold used. Notice that for both $k = 100$ and $k = 1000$, the recall curves flatten after a threshold of $\tau = 0.6$, and for all datasets they are essentially flat after $\tau \geqslant 0.7$. Thus, for such thresholds we incur essentially no quality loss by performing ball carving. Based on these empirical results, we choose the value of $\tau = 0.7$ in our end-to-end experiments.

In (§C.3), we show the impact on end-to-end QPS of ball carving on the MS MARCO dataset. For sequential re-ranking, we find that ball carving at a $\tau = 0.7$ threshold provides a $25\%$ QPS improvement, and when re-ranking is being done in parallel (over all cores simultaneously) it yields a $20\%$ QPS improvement. Moreover, with a threshold of $\tau = 0.7$, there were an average of 5.9 clusters created per query on MS MARCO. This reduces the number of query embeddings by $5.4\times$, down from the initial fixed setting of $|Q| = 32$. This finding shows that pre-clustering the queries before re-ranking gives non-trivial runtime improvements with negligible quality loss. It also suggests that a fixed setting of $|Q| = 32$ query embeddings used by existing approaches is likely excessive for MV similarity quality, and that fewer queries could achieve a similar performance.

**QPS vs. Recall.** A useful metric for retrieval is the number of *queries per second (QPS)* a system can serve at a given recall; evaluating the QPS of a system tries to fully utilize the system resources (e.g., the bandwidth of multiple memory channels and caches), and deployments where machines serve many queries simultaneously. Figure 7 shows the QPS vs. Recall@100 for MUVERA on a subset of the BEIR datasets, using different PQ schemes over the FDEs. We show results for additional datasets, as well as Recall@1000, in the Appendix. Using PQ-256-8 not only reduces the space usage of the FDEs by $32\times$, but also improves the QPS at the same query beamwidth by up to $20\times$, while incurring a minimal loss in end-to-end recall. Our method has a relatively small dependence on the dataset size, which is consistent with prior studies on graph-based ANNS data structures, since the number of distance comparisons made during beam search grows roughly logarithmically with increasing dataset size [25, 38]. We tried to include QPS numbers for PLAID [43], but unfortunately their implementation does not support running multiple queries in parallel, and is optimized for measuring latency.

**Latency and Recall Results vs. PLAID [43]** We evaluated MUVERA and PLAID [43] on the 6 datasets from the BEIR benchmark described earlier in (§3); Figure 8 shows that MUVERA achieves essentially equivalent Recall@$k$ as PLAID (within 0.4%) on MS MARCO, while obtaining up to $1.56\times$ higher recall on other datasets (e.g. HotpotQA). We ran PLAID using the recommended settings for their system, which reproduced their recall results for MS MARCO. Compared with PLAID, on

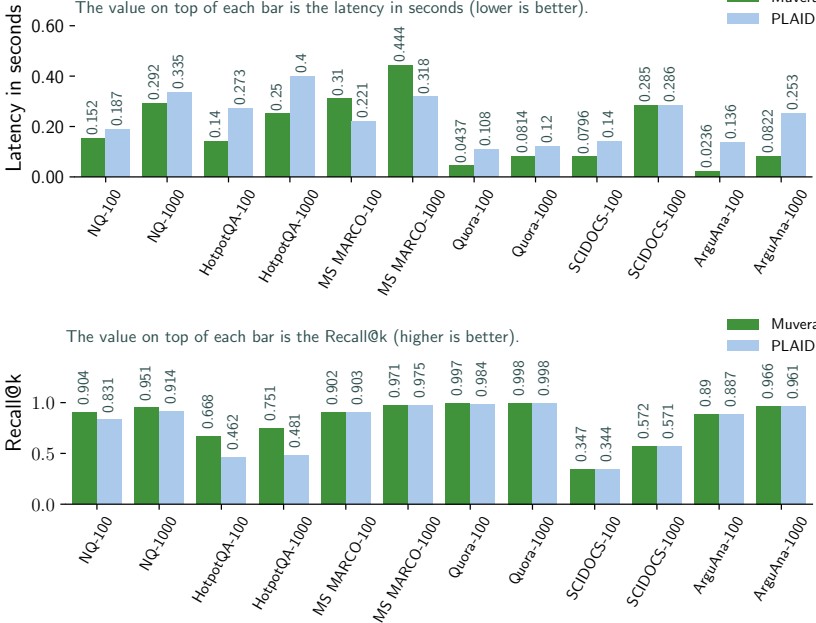

Figure 8: Bar plots showing the latency and Recall@$k$ of MUVERA vs PLAID on a subset of the BEIR datasets. The x-tick labels are formatted as dataset-$k$, i.e., optimizing for Recall@$k$ on the given dataset.

average over all 6 datasets and $k \in \{100, 1000\}$, MUVERA achieves 10% higher Recall@$k$ (up to 56% higher), and 90% lower latency (up to $5.7\times$ lower).

Importantly, MUVERA has consistently high recall and low latency across all of the datasets that we measure, and our method *does not* require costly parameter tuning to achieve this—all of our results use the same 10240-dimensional FDEs that are compressed using PQ with PQ-256-8; the only tuning in our system was to pick the first query beam-width over the $k$ that we rerank to that obtained recall matching that of PLAID. As Figure 8 shows, in cases like NQ and HotpotQA, MUVERA obtains much higher recall while obtaining lower latency. Given these results, we believe a distinguishing feature of MUVERA compared to prior multi-vector retrieval systems is that it achieves consistently high recall and low latency across a wide variety of datasets with minimal tuning effort.

## 4    Conclusion

In this paper, we presented MUVERA: a principled and practical MV retrieval algorithm which reduces MV similarity to SV similarity by constructing Fixed Dimensional Encoding (FDEs) of a MV representation. We prove that FDE dot products give high-quality approximations to Chamfer similarity (§2.1). Experimentally, we show that FDEs are a much more effective proxy for MV similarity, since they require retrieving 2-4× fewer candidates to achieve the same recall as the SV Heuristic (§3.1). We complement these results with an end-to-end evaluation of MUVERA, showing that it achieves an average of 10% improved recall with 90% lower latency compared with PLAID. Moreover, despite the extensive optimizations made by PLAID to the SV Heuristic, we still achieve significantly better latency on 5 out of 6 BEIR datasets we consider (§3). Given their retrieval efficiency compared to the SV heuristic, we believe that there are still significant gains to be obtained by optimizing the FDE method, and leave further exploration of this to future work.

**Broader Impacts and Limitations:**   While retrieval is an important component of LLMs, which themselves have broader societal impacts, these impacts are unlikely to result from our retrieval algorithm.  Our contribution simply improves the efficiency of retrieval, without enabling any fundamentally new capabilities.  As for limitations, while we outperformed PLAID, sometimes significantly, on 5 out of the 6 datasets we studied, we did not outperform PLAID on MS MARCO, possibly due to their system having been carefully tuned for MS MARCO given its prevalence. Additionally, we did not study the effect that the average number of embeddings $m_{avg}$ per document has on retrieval quality of FDEs; this is an interesting direction for future work.

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

# A    Missing Proofs from Section 2.1

In this section, we provide the missing proofs in Section 2.1. For convenience, we also reproduce theorem statements as they appear in the main text before the proofs. We begin by analyzing the runtime to compute query and document FDEs, as well as the sparsity of the queries.

**Lemma A.1.** *For any FDE parameters $k_{sim}, d_{proj}, R_{reps} \geqslant$ and sets $Q, P \subset \mathbb{R}^d$, we can compute $\mathbf{F}_q(Q)$ in time $T_q := O(R_{reps}|Q|d(d_{proj} + k_{sim}))$, and $\mathbf{F}_q(P)$ in time $O(T_q + R_{reps}|P|2^{k_{sim}}k_{sim})$. Moreover, $\mathbf{F}_q(Q)$ has at most $O(|Q|d_{proj}R_{reps})$ non-zero entries.*

*Proof.* We first consider the queries. To generate the queries, we must first project each of the $|Q|$ queries via the inner random linear productions $\psi_i : \mathbb{R}^d \to \mathbb{R}^{d_{\text{proj}}}$, which requires $O(|Q|dd_{\text{proj}}R_{\text{reps}})$ time to perform the matrix-query products for all repetitions. Next, we must compute $\varphi_i(q)$ for each $q \in Q$ and repetition $i \in [R_{\text{reps}}]$, Each such value can be compute in $d \cdot k_{\text{sim}}$ time to multiply the $q \in \mathbb{R}^d$ by the $k_{\text{sim}}$ Gaussian vectors. Thus the total running time for this step is $O(R_{\text{reps}}|Q|dk_{\text{sim}})$. Finally, summing the relevant values into the FDE once $\varphi_i(q), \psi_i(q)$ are computed can be done in $O(|Q|d_{\text{proj}})$ time. For sparsity, note that only the coordinate blocks in the FDE corresponding to clusters $k$ in a repetition $i$ with at least one $q \in |Q|$ with $\varphi_i(q) = k$ are non-zero, and there can be at most $O(R_{\text{reps}}|Q|)$ of these blocks, each of which has $O(d_{\text{proj}})$ coordinates.

The document runtime is similar, except with the additional complexity required to carry out the `fill_empty_clusters` option. For each repetition, the runtime required to find the closest $p \in P$ to a give cluster $k$ is $O(|P| \cdot k_{\text{sim}})$, since we need to run over all $|p|$ values of $\varphi(p)$ and check how many bits disagree with $k$. Thus, the total runtime is $O(R_{\text{reps}}|P|Bk_{\text{sim}}) = O(R_{\text{reps}}|P|2^{k_{\text{sim}}}k_{\text{sim}})$.    □

In what follows, we will need the following standard fact that random projections approximately preserve dot products. The proof is relatively standard, and can be found in [2], or see results on approximate matrix product [52] for more general bounds.

**Fact A.2** ([2]). *Fix $\varepsilon, \delta > 0$. For any $d \geqslant 1$ and $x, y \in \mathbb{R}^d$, let $S \in \mathbb{R}^{t \times d}$ by a matrix of independent entries distributed uniformly over $\{1, -1\}$, where $t = O(1/\varepsilon^2 \cdot \log \delta^{-1})$. Then we have $\mathbb{E}[\langle Sx, Sy \rangle] = \langle x, y \rangle$, and moreover with probability at least $1 - \delta$ we have*

$$|\langle Sx, Sy \rangle - \langle x, y \rangle| \leqslant \varepsilon \|x\|_2 \|y\|_2$$

To anaylze the approximations of our FDEs, we begin by proving an upper bound on the value of the FDE dot product. In fact, we prove a stronger result: we show that our FDEs have the desirable property of being *one-sided estimators* – namely, they never overestimate the true Chamfer similarity. This is summarized in the following Lemma.

**Lemma A.3** (One-Sided Error Estimator). *Fix any sets $Q, P \subset \mathbb{R}^d$ of unit vectors with $|Q| + |P| = m$. Then if $d = d_{proj}$, we always have*

$$\frac{1}{|Q|} \langle \mathbf{F}_q(Q), \mathbf{F}_{doc}(P) \rangle \leqslant \text{NCHAMFER}(Q, P)$$

*Furthermore, for any $\delta > 0$, if we set $d_{proj} = O(\frac{1}{\varepsilon^2} \log(m/\delta))$, then we have $\frac{1}{|Q|} \langle \mathbf{F}_q(Q), \mathbf{F}_{doc}(P) \rangle \leqslant \text{NCHAMFER}(Q, P) + \varepsilon$ in expectation and with probability at least $1 - \delta$.*

*Proof.* First claim simply follows from the fact that the average of a subset of a set of numbers can't be bigger than the maximum number in that set. More formally, we have:

$$\frac{1}{|Q|}\langle \mathbf{F}_{\mathrm{q}}(Q), \mathbf{F}_{\mathrm{doc}}(P)\rangle = \frac{1}{|Q|}\sum_{k=1}^{B}\sum_{\substack{q\in Q\\ \varphi(q)=k}}\frac{1}{|P\cap\varphi^{-1}(k)|}\sum_{\substack{p\in P\\ \varphi(p)=k}}\langle q,p\rangle$$

$$\leqslant \frac{1}{|Q|}\sum_{k=1}^{B}\sum_{\substack{q\in Q\\ \varphi(q)=k}}\frac{1}{|P\cap\varphi^{-1}(k)|}\sum_{\substack{p\in P\\ \varphi(p)=k}}\max_{p'\in P}\langle q,p'\rangle \qquad (4)$$

$$= \frac{1}{|Q|}\sum_{k=1}^{B}\sum_{\substack{q\in Q\\ \varphi(q)=k}}\max_{p'\in p}\langle q,p\rangle = \mathrm{NCHAMFER}(Q,P)$$

Which completes the first part of the lemma. For the second part, to analyze the case of $d_{\mathrm{proj}} < d$, when inner random projections are used, by applying Fact A.2, firstly we have $\mathbb{E}\left[\langle\boldsymbol{\psi}(p),\boldsymbol{\psi}(q)\right] = \langle q,p\rangle$ for any $q\in Q, p\in P$, and secondly, after a union bound we over $|P|\cdot|Q|\leqslant m^2$ pairs, we have $\langle q,p\rangle = \langle\boldsymbol{\psi}(p),\boldsymbol{\psi}(q)\rangle \pm \varepsilon$ simultaneously for all $q\in Q, p\in P$, with probability $1-\delta$, for any constant $C > 1$. The second part of the Lemma then follows similarly as above. $\qquad\square$

We are now ready to give the proof of our main FDE approximation theorem.

**Theorem 2.1** (FDE Approximation). *Fix any $\varepsilon,\delta > 0$, and sets $Q, P\subset\mathbb{R}^d$ of unit vectors, and let $m = |Q| + |P|$. Then setting $k_{sim} = O\left(\frac{\log(m\delta^{-1})}{\varepsilon}\right)$, $d_{proj} = O\left(\frac{1}{\varepsilon^2}\log(\frac{m}{\varepsilon\delta})\right)$, $R_{reps} = 1$, so that $d_{FDE} = (m/\delta)^{O(1/\varepsilon)}$, we have*

$$\mathrm{NCHAMFER}(Q,P) - \varepsilon \leqslant \frac{1}{|Q|}\langle\mathbf{F}_q(Q),\mathbf{F}_{doc}(P)\rangle \leqslant \mathrm{NCHAMFER}(Q,P) + \varepsilon$$

*in expectation, and with probability at least $1 - \delta$.*

*Proof of Theorem 2.1.* The upper bound follows from Lemma A.3, so it will suffice to prove the lower bound. We first prove the result in the case when there are no random projections $\psi$, and remove this assumption at the end of the proof. Note that, by construction, $\mathbf{F}_{\mathrm{q}}$ is a linear mapping so that $\mathbf{F}_{\mathrm{q}}(Q) = \sum_{q\in Q}\mathbf{F}(q)$, thus

$$\langle\mathbf{F}_{\mathrm{q}}(Q),\mathbf{F}_{\mathrm{doc}}(P)\rangle = \sum_{q\in Q}\langle\mathbf{F}_{\mathrm{q}}(q),\mathbf{F}_{\mathrm{doc}}(P)\rangle$$

So it will suffice to prove that

$$\mathbf{Pr}\left[\langle\mathbf{F}_{\mathrm{q}}(q),\mathbf{F}_{\mathrm{doc}}(P)\rangle \geqslant \max_{p\in P}\langle q,p\rangle - \varepsilon\right] \geqslant 1 - \varepsilon\delta/|Q| \qquad (5)$$

for all $q\in Q$, since then, by a union bound 5 will hold for all over all $q\in Q$ with probability at least $1 - \varepsilon\delta$, in which case we will have

$$\frac{1}{|Q|}\langle\mathbf{F}_{\mathrm{q}}(Q),\mathbf{F}_{\mathrm{doc}}(P)\rangle \geqslant \frac{1}{|Q|}\sum_{q\in Q}\left(\max_{p\in P}\langle q,p\rangle - \varepsilon\right)$$

$$= \mathrm{NCHAMFER}(Q,P) - \varepsilon \qquad (6)$$

which will complete the theorem.

In what follows, for any $x, y\in\mathbb{R}^d$ let $\theta(x,y)\in[0,\pi]$ be the angle between $x, y$. Now fix any $q\in Q$, and let $p^* = \arg\max_{p\in P}\langle q,p\rangle$, and let $\theta^* = \theta(q,p^*)$. By construction, there always exists some set of points $S\subset P$ such that

$$\langle\mathbf{F}_{\mathrm{q}}(q),\mathbf{F}_{\mathrm{doc}}(P)\rangle = \left\langle q, \frac{1}{|S|}\sum_{p\in S}p\right\rangle$$

Moreover, the RHS of the above equation is always bounded by 1 in magnitude, since it is an average of dot products of normalized vectors $q, p \in \mathbb{S}^{d-1}$. In particular, there are two cases. In case **(A)** $S$ is the set of points $p$ with $\boldsymbol{\varphi}(p) = \boldsymbol{\varphi}(q)$, and in case **(B)** $S$ is the single point $\arg\min_{p \in P} \|\boldsymbol{\varphi}(p) - \boldsymbol{\varphi}(q)\|_0$, where $\|x - y\|_0$ denotes the hamming distance between any two bit-strings $x, y \in \{0,1\}^{k_{\mathrm{sim}}}$, and we are interpreting $\boldsymbol{\varphi}(p), \boldsymbol{\varphi}(q) \in \{0,1\}^{k_{\mathrm{sim}}}$ as such bit-strings. Also let $g_1, \dots, g_{k_{\mathrm{sim}}} \in \mathbb{R}^d$ be the random Gaussian vectors that were drawn to define the partition function $\boldsymbol{\varphi}$. To analyze $S$, we first prove the following:

**Claim A.4.** For any $q \in Q$ and $p \in P$, we have

$$\mathbf{Pr}\left[\left|\|\boldsymbol{\varphi}(p) - \boldsymbol{\varphi}(q)\|_0 - k_{\mathrm{sim}} \cdot \frac{\theta(q,p)}{\pi}\right| > \sqrt{\varepsilon}k_{\mathrm{sim}}\right] \leqslant \left(\frac{\varepsilon\delta}{m^2}\right)$$

*Proof.* Fix any such $p$, and for $i \in [k_{\mathrm{sim}}]$ let $Z_i$ be an indicator random variable that indicates the event that $\mathbf{1}(\langle g_i, p\rangle > 0) \neq \mathbf{1}(\langle g_i, q\rangle > 0)$. First then note that $\|\boldsymbol{\varphi}(p) - \boldsymbol{\varphi}(q)\|_0 = \sum_{i=1}^{k_{\mathrm{sim}}} Z_i$. Now by rotational invariance of Gaussians, for a Gaussian vector $g \in \mathbb{R}^d$ we have $\mathbf{Pr}\left[\mathbf{1}(\langle g, x\rangle > 0) \neq \mathbf{1}(\langle g, y\rangle > 0)\right] = \frac{\theta(x,y)}{\pi}$ for any two vectors $x, y \in \mathbb{R}^d$. It follows that $Z_i$ is a Bernoulli random variable with $\mathbb{E}[Z_i] = \frac{\theta(x,y)}{\pi}$. By a simple application of Hoeffding's inequality, we have

$$\mathbf{Pr}\left[\left|\|\boldsymbol{\varphi}(p) - \boldsymbol{\varphi}(q)\|_0 - k_{\mathrm{sim}} \cdot \frac{\theta(q,p)}{\pi}\right| > \sqrt{\varepsilon}k_{\mathrm{sim}}\right] = \mathbf{Pr}\left[\left|\sum_{i=1}^{k_{\mathrm{sim}}} Z_i - \mathbb{E}\left[\sum_{i=1}^{k_{\mathrm{sim}}} Z_i\right]\right| > \sqrt{\varepsilon}k_{\mathrm{sim}}\right]$$
$$\leqslant \exp\left(-2\varepsilon k_{\mathrm{sim}}\right)$$
$$\leqslant \left(\frac{\varepsilon\delta}{m^2}\right)$$

$$(7)$$

where we took $k_{\mathrm{sim}} \geqslant 1/2 \cdot \log(\frac{m^2}{\varepsilon\delta})/\varepsilon$, which completes the proof. $\qquad\square$

We now condition on the event in Claim A.4 occurring for all $p \in P$, which holds with probability at least $1 - |P| \cdot \left(\frac{\varepsilon\delta}{m^2}\right) > 1 - \left(\frac{\varepsilon\delta}{m}\right)$ by a union bound. Call this event $\mathcal{E}$, and condition on it in what follows.

Now first suppose that we are in case **(B)**, and the set $S$ of points which map to the cluster $\boldsymbol{\varphi}(q)$ is given by $S = \{p'\}$ where $p' = \arg\min_{p \in P} \|\boldsymbol{\varphi}(p) - \boldsymbol{\varphi}(q)\|_0$. Firstly, if $p' = p^*$, then we are done as $\langle \mathbf{F}_{\mathrm{q}}(q), \mathbf{F}_{\mathrm{doc}}(P)\rangle = \langle q, p^*\rangle$, and 5 follows. Otherwise, by Claim A.4 we must have had $|\theta(q, p') - \theta(q, p^*)| \leqslant \pi \cdot \sqrt{\varepsilon}$. Using that the Taylor expansion of cosine is $\cos(x) = 1 - x^2/2 + O(x^4)$, we have

$$|\cos(\theta(q, p')) - \cos(\theta(q, p^*))| \leqslant O(\varepsilon)$$

Thus

$$\langle \mathbf{F}_{\mathrm{q}}(q), \mathbf{F}_{\mathrm{doc}}(P)\rangle = \langle q, p'\rangle$$
$$= \cos(\theta(q, p'))$$
$$\geqslant \cos(\theta(q, p^*)) - O(\varepsilon)$$
$$= \max_{p \in P}\langle q, p\rangle - O(\varepsilon)$$

$$(8)$$

which proves the desired statement 5 after a constant factor rescaling of $\varepsilon$.

Next, suppose we are in case **(A)** where $S = \{p \in P' \mid \boldsymbol{\varphi}(p) = \boldsymbol{\varphi}(q)\}$ is non-empty. In this case, $S$ consists of the set of points $p$ with $\|\boldsymbol{\varphi}(p) - \boldsymbol{\varphi}(q)\|_0 = 0$. From this, it follows again by Claim A.4

that $\theta(q, p) \leqslant \sqrt{\varepsilon}\pi$ for any $p \in S$. Thus, by the same reasoning as above, we have

$$
\begin{aligned}
\langle \mathbf{F}_\mathrm{q}(q), \mathbf{F}_\mathrm{doc}(P) \rangle &= \frac{1}{|S|} \sum_{p \in S} \cos(\theta(q, p')) \\
&\geqslant \frac{1}{|S|} \sum_{p \in S} (1 - O(\varepsilon)) \\
&\geqslant \frac{1}{|S|} \sum_{p \in S} (\langle q, p^* \rangle - O(\varepsilon)) \\
&= \max_{p \in P} \langle q, p \rangle - O(\varepsilon)
\end{aligned}
\tag{9}
$$

which again proves the desired statement 5 in case **(A)**, thereby completing the full proof in the case where there are no random projections.

To analyze the expectation, note that using the fact that $|\langle \mathbf{F}_\mathrm{q}(q), \mathbf{F}_\mathrm{doc}(P) \rangle| \leqslant 1$ deterministically, the small $O(\varepsilon\delta)$ probability of failure (i.e. the event that $\mathcal{E}$ does not hold) above can introduce at most a $O(\varepsilon\delta) \leqslant \varepsilon$ additive error into the expectation, which is acceptable after a constant factor rescaling of $\varepsilon$.

Finally, to incorporate projections, by standard consequences of the Johnson Lindenstrauss Lemma (Fact A.2) setting $d_{\mathrm{proj}} = O(\frac{1}{\varepsilon^2} \log \frac{m}{\varepsilon})$ and projecting via a random Gaussian or $\pm 1$ matrix from $\psi : \mathbb{R}^d \to \mathbb{R}^{d_{\mathrm{proj}}}$, for any set $S \subset P$ we have that $\mathbb{E}\left[ \langle \psi(q), \psi(\frac{1}{|S|} \sum_{p \in S} p) \rangle \right] = \langle q, \frac{1}{|S|} \sum_{p \in S} p \rangle$, and moreover that $\langle q, \frac{1}{|S|} \sum_{p \in S} p \rangle = \langle \psi(q), \psi(\frac{1}{|S|} \sum_{p \in S} p) \rangle \|q\|_2 \| \frac{1}{|S|} \sum_{p \in S} p \|_2 \pm \varepsilon$ for all $q \in Q, p \in P$ with probability at least $1 - \varepsilon\delta$. Note that $\|q\|_2 = 1$, and by triangle inequality $\| \frac{1}{|S|} \sum_{p \in S} p \|_2 \leqslant \frac{1}{|S|} \sum_{p \in S} \|p\|_2 = 1$. Thus, letting $\mathbf{F}_\mathrm{q}(Q), \mathbf{F}_\mathrm{doc}(P)$ be the FDE values without the inner projection $\psi$ and $\mathbf{F}_\mathrm{q}^{\boldsymbol{\psi}}(Q), \mathbf{F}_\mathrm{doc}^{\boldsymbol{\psi}}(P)$ be the FDE values with the inner projection $\psi$, conditioned on the above it follows that

$$
\begin{aligned}
\frac{1}{|Q|} \langle \mathbf{F}_\mathrm{q}^{\boldsymbol{\psi}}(Q), \mathbf{F}_\mathrm{doc}^{\boldsymbol{\psi}}(P) \rangle &= \frac{1}{|Q|} \sum_{q \in Q} \langle \mathbf{F}_\mathrm{q}^{\boldsymbol{\psi}}(q), \mathbf{F}_\mathrm{doc}^{\boldsymbol{\psi}}(P) \rangle \\
&= \frac{1}{|Q|} \sum_{q \in Q} (\langle \mathbf{F}_\mathrm{q}(q), \mathbf{F}_\mathrm{doc}(P) \rangle \pm \varepsilon) \\
&= \frac{1}{|Q|} \langle \mathbf{F}_\mathrm{q}(Q), \mathbf{F}_\mathrm{doc}(P) \rangle \pm \varepsilon
\end{aligned}
\tag{10}
$$

Finally, to analyze the expectation, note that since

$$
\left| \frac{1}{|Q|} \langle \mathbf{F}_\mathrm{q}(Q), \mathbf{F}_\mathrm{doc}(P) \rangle \right| \leqslant \frac{1}{|Q|} \sum_{q \in Q} |\langle \mathbf{F}_\mathrm{q}(q), \mathbf{F}_\mathrm{doc}(P) \rangle| \leqslant 1
$$

as before conditioning on this small probability event changes the expectation of 5 by at most a $\varepsilon$ additive factor, which completes the proof of the Theorem after a constant factor rescaling of $\varepsilon$.

$\square$

Equipped with Theorem 2.1, as well as the sparsity bounds from Lemma A.1, we are now prepared to prove our main theorem on approximate nearest neighbor search under the Chamfer Similarity.

**Theorem 2.2.** *Fix any $\varepsilon > 0$, query $Q$, and dataset $P = \{P_1, \dots, P_n\}$, where $Q \subset \mathbb{R}^d$ and each $P_i \subset \mathbb{R}^d$ is a set of unit vectors. Let $m = |Q| + \max_{i \in [n]} |P_i|$. Then setting $k_{\boldsymbol{sim}} = O(\frac{\log m}{\varepsilon})$, $d_{\boldsymbol{proj}} = O(\frac{1}{\varepsilon^2} \log(m/\varepsilon))$ and $R_{\boldsymbol{reps}} = O(\frac{1}{\varepsilon^2} \log n)$ so that $d_{FDE} = m^{O(1/\varepsilon)} \cdot \log n$. Then setting $i^* = \arg\max_{i \in [n]} \langle \mathbf{F}_q(Q), \mathbf{F}_{doc}(P_i) \rangle$, with high probability (i.e. $1 - 1/\operatorname{poly}(n)$) we have:*

$$
\mathrm{NCHAMFER}(Q, P_{i^*}) \geqslant \max_{i \in [n]} \mathrm{NCHAMFER}(Q, P_i) - \varepsilon
$$

*Given the query $Q$, the document $P^*$ can be recovered in time $O\left(|Q| \max\{d, n\} \frac{1}{\varepsilon^4} \log(\frac{m}{\varepsilon}) \log n \right)$.*

*Proof of Theorem 2.2.* First note, for a single repetition, for any subset $P_j \in D$, by Theorem 2.1 we have

$$\mathbb{E}\left[\langle \mathbf{F}_q(Q), \mathbf{F}_{\text{doc}}(P_j)\rangle\right] = \text{NCHAMFER}(Q, P) \pm \varepsilon$$

Moreover, as demonsrated in the proof of Theorem 2.1, setting $\delta = 1/10$, we have

$$\left|\frac{1}{|Q|}\langle \mathbf{F}_q(Q), \mathbf{F}_{\text{doc}}(P_j)\rangle\right| \leqslant \frac{1}{|Q|}\sum_{q \in Q}|\langle \mathbf{F}_q(q), \mathbf{F}_{\text{doc}}(P_j)\rangle| \leqslant 1$$

It follows that for each repetition $i \in [R_{\text{reps}}]$, letting $\mathbf{F}_q(Q)^i, \mathbf{F}_{\text{doc}}(P_j)^i$ be the coordinates in the final FDE vectors corresponding to that repetition, the random variable $X_i = \frac{1}{|Q|}\langle \mathbf{F}_q^i(Q), \mathbf{F}_{\text{doc}}^i(P_j)\rangle$ is bounded in $[-1, 1]$ and has expectation $\text{NCHAMFER}(Q, P_j) \pm \varepsilon$. By Chernoff bounds, averaging over $R_{\text{reps}} = O(\frac{1}{\varepsilon^2}\log(n))$ repetitions, we have

$$\left|\sum_{i=1}^{R_{\text{reps}}}\frac{1}{R_{\text{reps}}|Q|}\langle \mathbf{F}_q^i(Q), \mathbf{F}_{\text{doc}}^i(P_j)\rangle - \text{NCHAMFER}(Q, P_j)\right| \leqslant 2\varepsilon \tag{11}$$

with probability $1 - 1/n^C$ for any arbitrarily large constant $C > 1$. Note also that $\sum_{i=1}^{R_{\text{reps}}}\frac{1}{R_{\text{reps}}|Q|}\langle \mathbf{F}_q^i(Q), \mathbf{F}_{\text{doc}}^i(P_j)\rangle = \frac{1}{R_{\text{reps}}|Q|}\langle \mathbf{F}_q(Q), \mathbf{F}_{\text{doc}}(P_j)\rangle$, where $\mathbf{F}_q(Q), \mathbf{F}_{\text{doc}}(P_j)$ are the final FDEs. We can then condition on (11) holding for all documents $j \in [n]$, which holds with probability with probability $1 - 1/n^{C-1}$ by a union bound. Conditioned on this, we have

$$\text{NCHAMFER}(Q, P_{i^*}) \geqslant \frac{1}{R_{\text{reps}}|Q|}\langle \mathbf{F}_q(Q), \mathbf{F}_{\text{doc}}(P_{i^*})\rangle - 2\varepsilon$$

$$= \max_{j \in [n]}\frac{1}{R_{\text{reps}}|Q|}\langle \mathbf{F}_q(Q), \mathbf{F}_{\text{doc}}(P_j)\rangle - 2\varepsilon \tag{12}$$

$$\geqslant \max_{j \in [n]}\text{NCHAMFER}(Q, P_j) - 6\varepsilon$$

which completes the proof of the approximation after a constant factor scaling of $\varepsilon$. The runtime bound follows from the runtime required to compute $\mathbf{F}_q(Q)$, which is $O(|Q|R_{\text{reps}}d(d_{\text{proj}} + k_{\text{sim}})) = O(|Q|\frac{\log n}{\varepsilon^2}d(\frac{1}{\varepsilon}\log(m/\varepsilon) + \frac{1}{\varepsilon}\log m)$, plus the runtime required to brute force search for the nearest dot product. Specifically, note that each of the $n$ FDE dot products can be computed in time proportional to the sparsity of $\mathbf{F}_q(Q)$, which is at most $O(|Q|d_{\text{proj}}R_{\text{reps}}) = O(|Q|\frac{1}{\varepsilon^4}\log(m/\varepsilon)\log n)$. Adding these two bounds together yields the desired runtime. $\square$

## B  Additional Dataset Information

In Table 9 we provide further dataset-specific information on the BEIR retrieval datasets used in this paper. Specifically, we state the sizes of the query and corpuses used, as well as the average number of embeddings produced by the ColBERTv2 model per document. Specifically, we consider the six BEIR retrieval datasets MS MARCO [40], NQ [31], HotpotQA [53], ArguAna [47], SciDocs [11], and Quora [46], Note that the MV corpus (after generating MV embeddings on all documents in a corpus) will have a total of #Corpus × (Avg # Embeddings per Doc) token embeddings. For even further details, see the BEIR paper [46].

|  | MS MARCO | HotpotQA | NQ | Quora | SciDocs | ArguAna |
|---|---|---|---|---|---|---|
| #Queries | 6,980 | 7,405 | 3,452 | 10,000 | 1,000 | 1,406 |
| #Corpus | 8.84M | 5.23M | 2.68M | 523K | 25.6K | 8.6K |
| Avg # Embeddings per Doc | 78.8 | 68.65 | 100.3 | 18.28 | 165.05 | 154.72 |

Figure 9: Dataset Specific Statistics for the BEIR datasets considered in this paper.

# C  Additional Experiments and Plots

In this Section, we provide additional plots to support the experimental results from Section 3. We provide plots for all six of the datasets and additional ranges of the $x$-axis for our experiments in Section (§3.1), as well as additional experimental results, such as an evaluation of variance, and of the quality of final projections in the FDEs.

**FDE vs. SV Heuristic Experiments.**  In Figures 10 and 11, we show further datasets and an expanded recall range for the comparison of the SV Heuristic to retrieval via FDEs. We find that our 4k+ dimensional FDE methods outperform even the deduplciated SV heuristic (whose cost is somewhat unrealistic, since the SV heuristic must over-retrieve to handle duplicates) on most datasets, especially in lower recall regimes. In Table 1, we compare how many candidates must be retrieved by the SV heuristic, both with and without the deduplication step, as well as by our FDE methods, in order to exceed a given recall threshold.

| Recall Threshold | SV non-dedup | SV dedup | 20k FDE | 10k FDE | 4k FDE | 2k FDE |
|---|---|---|---|---|---|---|
| 80% | 1200 | 300 | 60 | 60 | 80 | 200 |
| 85% | 2100 | 400 | 90 | 100 | 200 | 300 |
| 90% | 4500 | 800 | 200 | 200 | 300 | 800 |
| 95% | >10000 | 2100 | 700 | 800 | 1200 | 5600 |

Table 1: FDE retrieval vs SV Heuristic: number of candidates that must be retrieved by each method to exceed a given recall on MS MARCO. The first two columns are for the SV non-deduplicated and deduplicated heuristics, respectively, and the remaining four columns are for the FDE retrieved candidates with FDE dimensions $\{20480, 10240, 4096, 2048\}$, respectively. Recall@$N$ values were computed in increments of 10 between 10-100, and in increments of 100 between 100-10000, and were not computed above $N > 10000$.

**Retrieval quality with respect to exact Chamfer.**  In Figure 12, we display the full plots for FDE Recall with respects to recovering the 1-nearest neighbor under Chamfer Similarity for all six BEIR datasets that we consider, including the two omitted from the main text (namely, SciDocs and ArguAna).

## C.1  Variance of FDEs.

Since the FDE generation is a randomized process, one natural concern is whether there is large variance in the recall quality across different random seeds. Fortunately, we show that this is not the case, and the variance of the recall of FDE is essentially negligible, and can be easily accounted for via minor extra retrieval. To evaluate this, we chose four sets of FDE parameters $(R_{\text{reps}}, k_{\text{sim}}, d_{\text{proj}})$ which were Pareto optimal for their respective dimensionalities, generated 10 independent copies of the query and document FDEs for the entire MS MARCO dataset, and computed the average recall@100 and 1000 and standard deviation of these recalls. The results are shown in Table 2, where for all of the experiments the standard deviation was between $0.08$-$0.3\%$ of a recall point, compared to the $80$-$95\%$ range of recall values. Note that Recall@1000 had roughly twice as small standard deviation as Recall@100.

| FDE params $(R_{\text{reps}}, k_{\text{sim}}, d_{\text{proj}})$ | $(20, 5, 32)$ | $(20, 5, 16)$ | $(20, 4, 16)$ | $(20, 4, 8)$ |
|---|---|---|---|---|
| FDE Dimension | 20480 | 10240 | 5120 | 2560 |
| Recall@100 | 83.68 | 82.82 | 80.46 | 77.75 |
| Standard Deviation | 0.19 | 0.27 | 0.29 | 0.17 |
| Recall@1000 | 95.37 | 94.88 | 93.67 | 91.85 |
| Standard Deviation | 0.08 | 0.11 | 0.16 | 0.12 |

Table 2: Variance of FDE Recall Quality on MS MARCO.

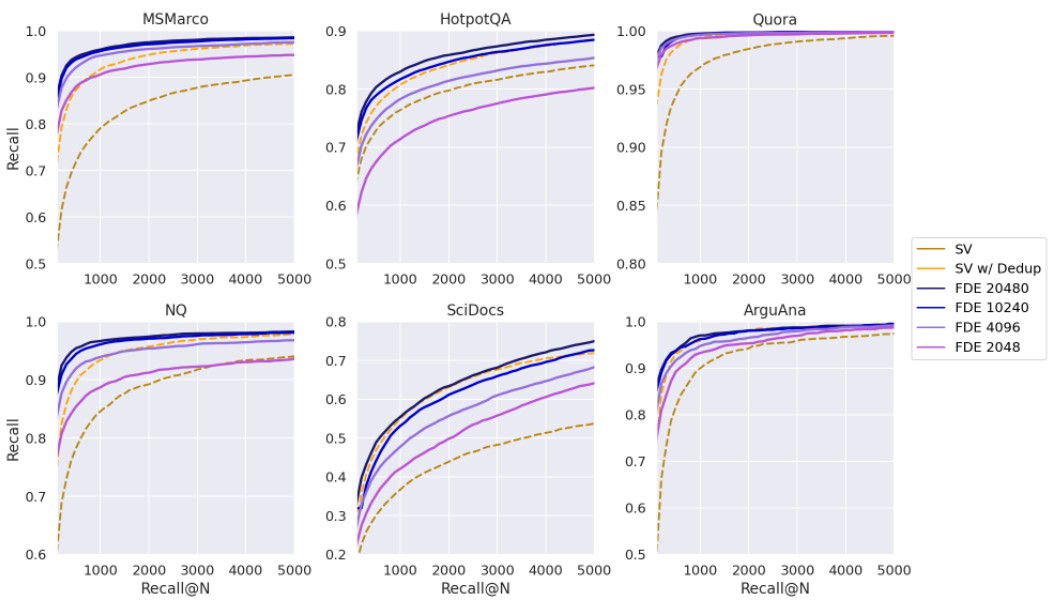

Figure 10: FDE retrieval vs SV Heuristic, Recall@100-5000

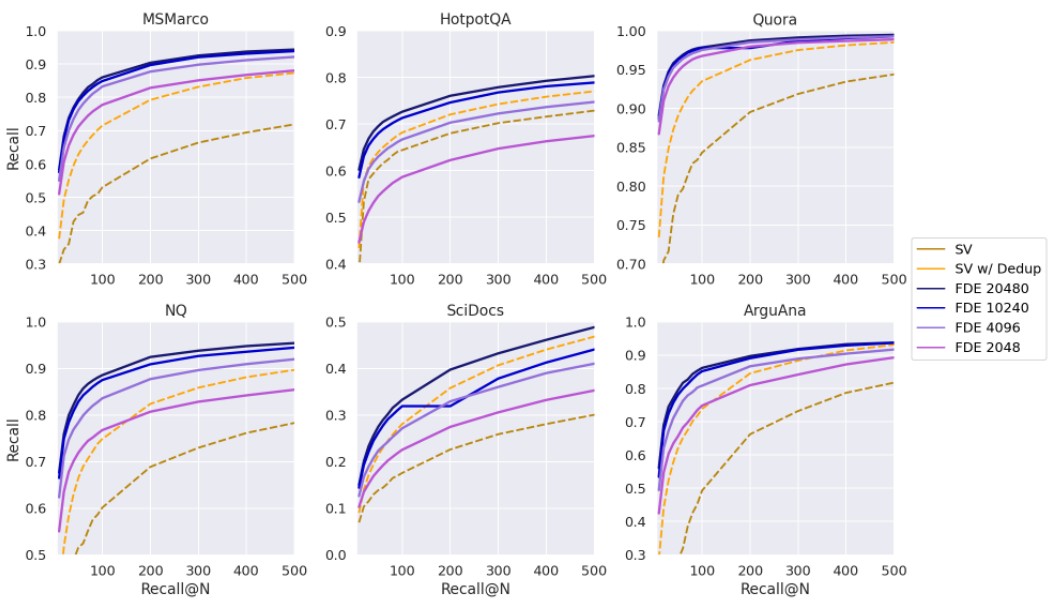

Figure 11: FDE retrieval vs SV Heuristic, Recall@5-500

| Experiment | w/o projection | w/ projection | w/o projection | w/ projection |
|---|---|---|---|---|
| Dimension | 2460 | 2460 | 5120 | 5120 |
| Recall@100 | 77.71 | 78.82 | 80.37 | 83.35 |
| Recall@1000 | 91.91 | 91.62 | 93.55 | 94.83 |
| Recall@10000 | 97.52 | 96.64 | 98.07 | 98.33 |

Table 3: Recall Quality of Final Projection based FDEs with $d_{\text{FDE}} \in \{2460, 5120\}$

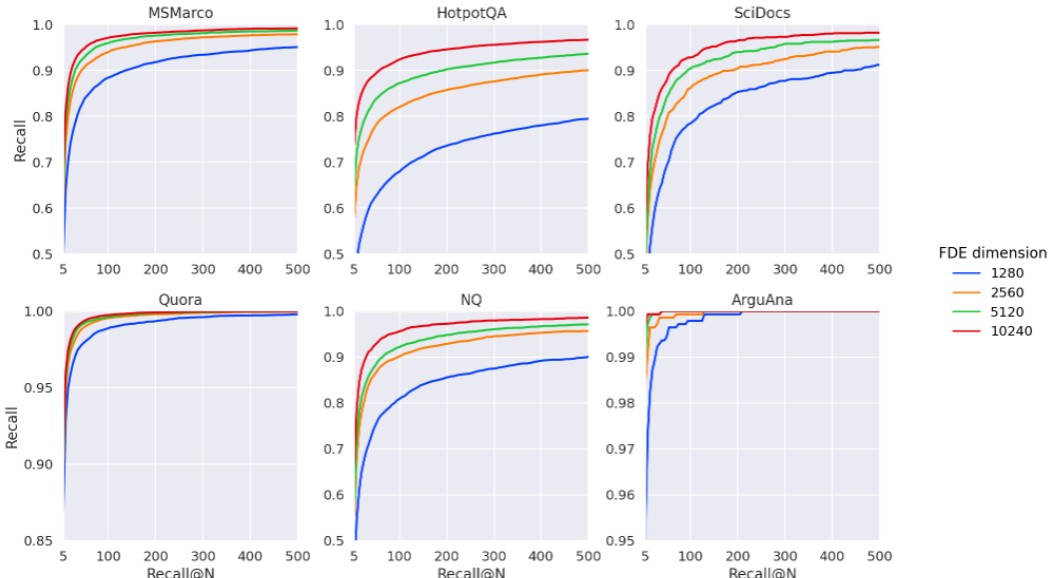

Figure 12: Comparison of FDE recall with respect to the most similar point under Chamfer.

| Experiment | w/o projection | w/ projection | w/o projection | w/ projection |
|---|---|---|---|---|
| Dimension | 10240 | 10240 | 20480 | 20480 |
| Recall@100 | 82.31 | 85.15 | 83.36 | 86.00 |
| Recall@1000 | 94.91 | 95.68 | 95.58 | 95.95 |
| Recall@10000 | 98.76 | 98.93 | 98.95 | 99.17 |

Table 4: Recall Quality of Final Projection based FDEs with $d_{\text{FDE}} \in \{10240, 20480\}$

## C.2 Comparison to Final Projections.

We now show the effect of employing final projections to reduce the target dimensionality of the FDE's. For all experiments, the final projection $\psi'$ is implemented in the same way as inner projections are: namely, via multiplication by a random $\pm 1$ matrix. We choose four target dimensions, $d_{\text{FDE}} \in \{2460, 5120, 10240, 20480\}$, and choose the Pareto optimal parameters $(R_{\text{reps}}, k_{\text{sim}}, d_{\text{proj}})$ from the grid search without final projections in Section 3.1, which are $(20, 4, 8), (20, 5, 8), (20, 5, 16), (20, 5, 32)$. We then build a large dimensional FDE with the parameters $(R_{\text{reps}}, k_{\text{sim}}, d_{\text{proj}}) = (40, 6, 128)$. Here, since $d = d_{\text{proj}}$, we do not use any inner projections when constructing the FDE. We then use a single random final projection to reduce the dimensionality of this FDE from $R_{\text{reps}} \cdot 2^{k_{\text{sim}}} \cdot d_{\text{proj}} = 327680$ down to each of the above target dimensions $d_{\text{FDE}}$. The results are show in Tables 3 and 4. Notice that incorporating final projections can have a non-trivial impact on recall, especially for Recall@100, where it can increase by around 3%. In particular, FDEs with the final projections are often better than FDEs with twice the dimensionality without final projections. The one exception is the 2460-dimensional FDE, where the Recall@100 only improved by 1.1%, and the Recall@1000 was actually lower bound 0.3%.

## C.3 Ball Carving

Continuing our discussion from Section 3.2, we show that ball-carving at this threshold of 0.7 gives non-trivial efficiency gains. Specifically, in Figure 13, we plot the per-core queries-per-second of re-ranking (i.e. computing $\text{CHAMFER}(Q_C, P)$) against varying ball carving thresholds for the MS MARCO dataset. Please see the discussion in Section 3.2 for analysis of the figure.

## C.4 Product Quantization

**PQ Details** We implemented our product quantizers using a simple "textbook" $k$-means based quantizer. Recall that AH-$C$-$G$ means that each consecutive group of $G$ dimensions is represented by $C$ centers. We train the quantizer by: (1) taking for each group of dimensions the coordinates

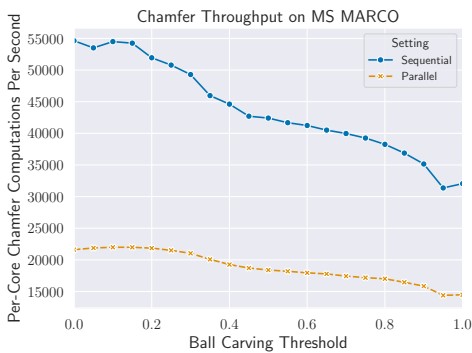

Figure 13: Per-Core Re-ranking QPS versus Ball Carving Threshold, on MS MARCO dataset.

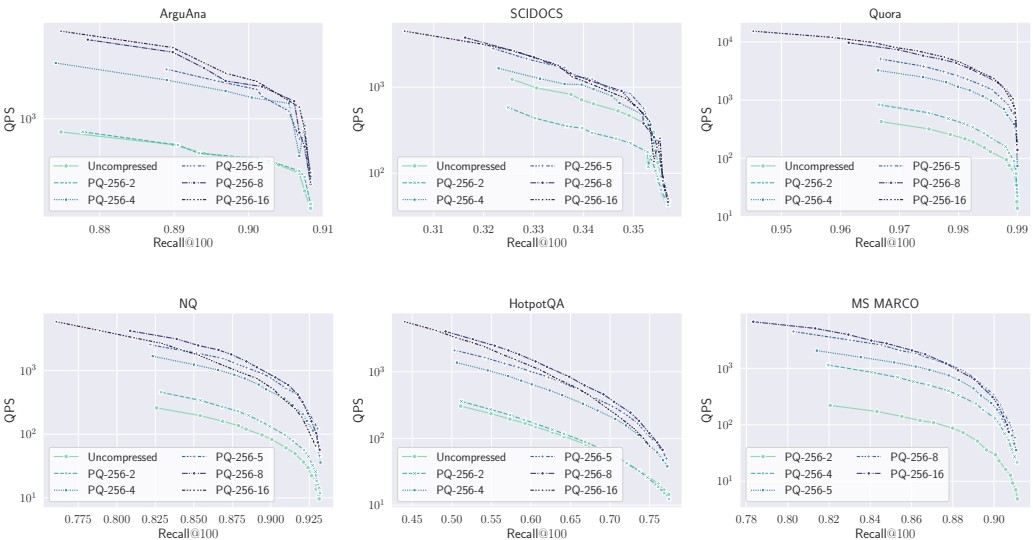

Figure 14: Plots showing the QPS vs. Recall@100 for MUVERA on the BEIR datasets we evaluate in this paper. The different curves are obtained by using different PQ methods on 10240-dimensional FDEs.

of a sample of at most 100,000 vectors from the dataset, and (2) running $k$-means on this sample using $k = C = 256$ centers until convergence. Given a vector $x \in \mathbb{R}^d$, we can split $x$ into $d/G$ blocks of coordinates $x_{(1)}, \ldots, x_{(d/G)} \in \mathbb{R}^G$ each of size $G$. The block $x_{(i)}$ can be compressed by representing $x_{(i)}$ by the index of the centroid from the $i$-th group that is nearest to $x_{(i)}$. Since there are 256 centroids per group, each block $x_{(i)}$ can then be represented by a single byte.

**Results** In Figures 14 and 15 we show the full set of results for our QPS experiments from Section 3.2 on all of the BEIR datasets that we evaluated in this paper. We include results for both Recall@100 (Figure 14) and Recall@1000 (Figure 15).

We find that PQ-256-8 is consistently the best performing PQ codec across all of the datasets that we tested. Not using PQ at all results in significantly worse results (worse by at least 5× compared to using PQ) at the same beam width for the beam; however, the recall loss due to using PQ-256-8 is minimal, and usually only a fraction of a percent. Since our retrieval engine works by over-retrieving with respect to the FDEs and then reranking using Chamfer similarity, the loss due to approximating the FDEs using PQ can be handled by simply over-retrieving slightly more candidates.

We also observe that the difference between different PQ codecs is much more pronounced in the lower-recall regime when searching for the top 1000 candidates for a query. For example, most of the plots in Figure 15 show significant stratification in the QPS achieved in lower recall regimes,

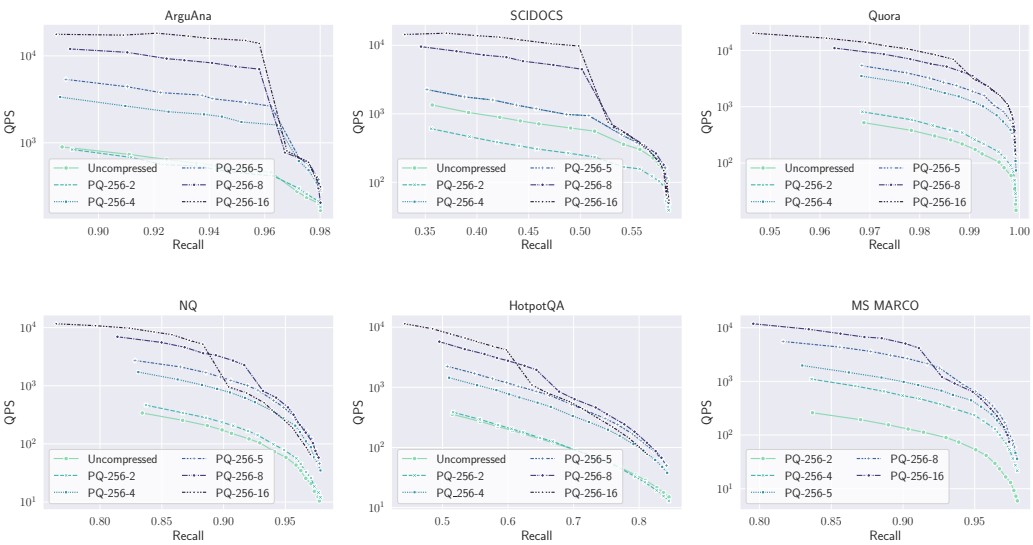

Figure 15: Plots showing the QPS vs. Recall@1000 for MUVERA on the BEIR datasets we evaluate in this paper. The different curves are obtained by using different PQ methods on 10240-dimensional FDEs.

with PQ-256-16 (the most compressed and memory-efficient format) usually outperforming all others; however, for achieving higher recall, PQ-256-16 actually does much worse than slightly less compressed formats like PQ-256-8 and PQ-256-4.

