# OpenReview forum: "MUVERA: Multi-Vector Retrieval via Fixed Dimensional Encoding"
_NeurIPS.cc/2024/Conference — NeurIPS 2024 poster_

### Official Review · Reviewer_Z5yi · 2024-07-02

**Soundness:** 2
**Presentation:** 3
**Contribution:** 3
**Rating:** 4
**Confidence:** 4

**Summary:**

The paper studies how to optimize multi-vector retrieval under the MaxSim similarity function (commonly used in NLP literatures like ColBERT). This is done by mapping queries and item embeddings asymmetrically to fixed dimensional embeddings (FDEs), such that in the mapped embedding space, the inner product similarity is a good approximation of the original MaxSim similarity. This approach then enables classical MIPS techniques to be applied to optimize for MaxSim similarity search. The authors compare FDEs with PLAID / SV heuristics, and found it to outperform those optimized similarity search baselines.

**Strengths:**

**S1**: I really like the MIPS approximation construction for MaxSim in Section 2. While the techniques used are not that novel (e.g., Section 2 heavily reminds me of the classical AND- and OR- constructions on top of LSH etc.), being able to reduce MaxSim to MIPS is likely of significant interest to many researchers in NLP and vector search community.

**S2**: Reducing MaxSim to MIPS enables many classical optimizations for MIPS to be applied incl PQ, RP (line 177-180) etc, likely significantly improving the usability and applicability of ColBERT-style approaches.

**S3**: MaxSim similarity (and related setting under late-interaction settings) is an important topic in NLP, as inner products have been proven to be suboptimal for similarity search, especially when we are interested in Recall@K for small Ks (e.g., 10s-50s).

**Weaknesses:**

**W1**: With FDE, the MaxSim approach (used in ColBERT and other work)'s effectiveness is degraded dramatically, to the point that it underperforms dense retrieval baselines.  E.g., Figure 3 (LHS) suggest that Recall@100 of FDE is quite bad vs Exact MaxSim in ColBERT (up to ~.84, vs ~.92). The main issue here is that, gains of non-MIPS similarity approaches (like MaxSim and more recently DSI/NCI etc.) are primarily on smaller @Ks (e.g., ColBERT v2's quality gains are primarily demonstrated on Metric@10 in their original paper). The current paper only reported Recall@K={100, 1K, 10K}, which are not aligned with intended use cases for ColBERT (as it didn't provide that much gains over dense baselines for these Ks).
- Even at 100, the 1-.84/.92 = 0.087 degradation in recall will likely make MaxSim+FDE underperform classical dense retrieval baselines. E.g., on MS MACRO, DPR achieves 82.2 R@50 whereas ColBERT achieves 82.9 R@50 per [1] (and gap @100 will be smaller). So adjusting for FDE's quality degradation, we get 82.2 for dual encoder baselines but just 82.9 * .84/.92 = 75.69 for MaxSim, and people will ask - why not use classical dense encoders like DPRs to begin with?
- Additionally, to achieve the ~.84 recall, FDE encodings already used 20K floats vs 10,087 in MS MACRO multi-vector baselines per Fig. 3...

**W2**: Random projection techniques (incl. their multi-hash variants, or $R_{reps}$ in this paper), and production quantization techniques used in FDEs are commonly used in MIPS systems, and generally universally reduce memory access costs and improve throughput/latency, etc. of nearest neighbor methods (after hparam tuning). Have we evaluated applying random projection and PQ techniques directly to baselines like ColBERTv2, PLAID, etc.? How much of the throughput gains presented in the paper are due to RPs/PQs?

**W3**: Writing could be improved.
- It would be good to add a table of notations, at least to appendix to help with references. Even though I work on vector search (incl multi-vector search and variants), I ended up spending quite some time figuring out what the authors meant by various notations in Section 2.
- Line 141-142: "To resolve this, we set ~p(k) to be the centroid of the p ∈ P’s with ϕ(p) = ϕ(q)". - This sentence is confusing as the FDE mapping for query (p) should not be item (q) aware. Please consider rewriting it to reflect the actual mapping done in Equation (3).
- Section 3: consider removing codebase/dataset licenses (CC BY-SA MIT etc.) as most readers may not find them useful.
- Line 311 - "We run latency experiments using a single thread, and run our QPS experiments on all 176 threads."  Could you explain why latency experiments are run with a single thread? As using additional threads should further improve latency.
- Typos: Line 132-133 are more are more likely to land in -> are more likely to land in


**References**:
- [1] Ren et al. PAIR: Leveraging Passage-Centric Similarity Relation for Improving Dense Passage Retrieval. ACL 2021

**Questions:**

**Q1**: Would it be possible to report effectiveness of FDE + MaxSim vs corresponding baselines (incl dense retrieval and ColBERTv2 etc.), esp. for common Ks (10, 50, 100)?

**Q2**: Can we equally apply techniques like PQ and random projection to baselines (again ideally dense retrieval with a comparable embedding dimensionality to normalize I/O costs), and evaluate how much throughput gains are due to PQs/RPs vs the particular construction of FDEs proposed in this paper?

**Q3**: FDE's approximation quality seems to heavily rely on how the B clusters are defined. If B is large: a) the overall dimensionality is large, requiring significantly more memory accesses, and b) Max(P, Q) may not map to anything.  OTOH if B is small, many false positives collisions may occur, resulting in (potentially significant) overestimation of the MaxSim similarity function. Is there a reliable way of grid searching B (other than in 2^k as done in Sec 3.1)? Could we discuss this somewhere in the paper?

---

> ### Author Rebuttal · Authors · 2024-08-05
>
> We thank the reviewer for their detailed comments and suggestions. We reply to the main questions and concerns below.
>
> > W1:
>
> We first note that the FDE technique, like the SV Heuristic, is an approach for multi-vector retrieval that must be followed by a re-ranking step with the exact Chamfer similarity. Therefore, the raw recall of the FDE method should not be compared to the brute-force ColbertV2 recall (or to other dense retrieval baselines), but rather to other alternative methods for MV over-retrieval. The primary such alternative (used in PLAID and others) is the SV Heuristic, and we show that FDE’s are significantly more efficient than this method by a factor of 2.6-5x. The message of Figure 3 is that by using a (e.g.) 2.5k dimensional FDE, retrieving 1000 candidates and reranking, one can obtain near-optimal recall@100 with respect to the brute force Chamfer (see Figure 3, middle, at the point where the FDE pareto curve passes the red “Exact Chamfer R@100” line). In contrast, the SV heuristic needs to retrieve many more (2k-5k) candidates to achieve the same recall after reranking.
>
> > “MaxSim+FDE underperform dense retrieval baselines.”
>
> We emphasize again that our model is *not* returning the 100 candidates retrieved by the FDE. Instead, it would retrieve more candidates, say 500 or 1000, rerank them with Chamfer, and output the top-100 after reranking. This gives us significantly better R@100 than 0.84, for instance, in Figure 7, in our end-to-end evaluation, we achieve .902 R@100 and 0.971 R@1000 on MS Marco, both of which are much closer to the brute-force ColBERTv2 recall (within 1-2%). Also note that this matches the recall obtained by PLAID in those experiments. Also note that PLAID (which uses the SV heuristic for initial retrieval) must also perform the same over-retrieval and reranking steps.
>
> > “Additionally, to achieve the ~.84 recall, FDE encodings already used 20K floats vs 10,087 in MS MACRO multi-vector baselines…”
>
> The same above point about reranking applies to this comment as well. Specifically, if we wanted to achieve end-to-end 0.84 recall, we would not use the raw output of the 20k dimensional FDE’s. Instead, we could use the (e.g.) 960 dimensional FDE, which attains .855 R@1000 (shown in Figure 3), retrieve 1000 candidates, and rerank to output the top 100. This FDE would then be 10x smaller than the 10k floats used in the MV baseline. Alternatively, we could use our 4k dimensional FDE’s, which achieve 0.88 Recall@500, and retrieve 500 and re-rank to get roughly the same result (if not better).
>
> With regards to the choice of k used in our Recall@k experiments, we point out that the ColBERT paper also studied similar values of k, namely (50,200,1000), as well as the PLAID paper (100,1k). Since 100,1k were the metrics considered by PLAID, our primary baseline, our end-to-end recall results focused on the same values, although we would be happy to add additional points of comparison at low recall ranges.
>
> > W2 + Q2:
>
> With regards to using product quantization in baselines like PLAID or ColBERTv2, we point out that those papers *extensively* utilize quantization methods in their algorithm already. Specifically, the ColBERTv2 retriever compresses points by representing each point by a nearby centroid plus a compressed residual (1-2 bits per coordinate), where the centroids are sampled from the dataset in pre-processing. PLAID, which is built on top of ColBERTv2, also uses the same compression scheme with additional optimizations. Thus, these methods crucially use complex and highly tailored quantization methods. One of our main contributions is to show that an out-of-the-box textbook product quantization method can be used to achieve similar results when applied to FDE’s.
>
> > “how much throughput gains are due to PQs/RPs vs the particular construction of FDEs proposed in this paper?”
>
> In Figures 14 and 15 we address exactly this question, by plotting Recall vs QPS curves for different datasets and different PQ methods, including uncompressed methods. The figure shows that using PQ gives significant throughput gains, which are needed to be competitive with PLAID (which, as discussed above, also extensively uses quantization techniques).
>
> > W3:
>
> We greatly appreciate the suggestions for improving the writing, and will be sure to incorporate them.
>
> > Q1:
>
> As discussed above, we compare FDE’s without any reranking brute force ColBERTv2 in Figure 3, for k=100 and 1000, as these were the Recall@K metrics considered in the PLAID paper. We will add additional experiments for smaller k to the paper as well. Since our work focuses on designing a standalone retrieval algorithm for MV models, we do not compare to IR methods (such as SV dense retrievers).
>
> > Q3:
>
> This is a fantastic question, and addressing it is a key contribution of our work. Specifically, we introduce the method “fill_empty_clusters” on Page 5 (see 166-180) to solve this problem, which ensures that at least one document vector maps to every cluster. As a result, there are no empty clusters on the document side, so there can be no “misses”. This makes tuning B easy – increasing B will always improve performance, so increase it as large as possible within your dimension constraints. This is also illustrated in the grid search experiments, where increasing B improves performance. We will emphasize this further by referring back to the paragraph on Page 5 in the section on grid searching.
>
> > “... (potentially significant) overestimation of the MaxSim similarity function.”
>
> It is certainly true that setting B too small could result in collisions, but we note that this could actually only result in an underestimation. In fact, we prove that our method never over-estimates (Lemma A.3), since every query vector will get matched to some document vector (or none, if fill_empty_clusters is not enabled) which can only result in an underestimate of the best similarity.

---

> > ### Comment · Reviewer_Z5yi · 2024-08-08
> > **some additional questions**
> >
> > Thanks for providing the clarifications. I have some additional questions based on the authors' responses:
> >
> > * inference setups used w/ FDE + MaxSim: thanks for your response to W1 as I was confused about the exact inference setup earlier. polishing writing further with a paragraph highlighting FDE's inference setup somewhere would be useful. Some comments:
> >   * ColBERTv2 and PLAID are more *multi-vector* SV baselines, whereas the inference setup in this paper differs by doing lightweight prefiltering w/ FDE SV to a larger set. This reminds me of retrieve-then-rerank in IR which is a reasonably common setup and may need discussions/comparisons. Some related work in this area might include Learning Query-dependent Prefilters for Scalable Image Retrieval. CVPR'09 [1], Revisiting Neural Retrieval on Accelerators. KDD'23 [2] etc. In particular [2] also proposed to retrieve with learned single-vector SV followed by re-scoring with learned similarities. It might be useful to compare learned SVs which seem to work for small ds (64d in [2]) vs FDE constructed SVs (2k-20k dimension here, albeit different datasets etc)?
> >    * writing: would using "multi-vector heuristics" to characterize the method in ColBERTv2/PLAID make more sense, given prior work have explored two-stage SV for learned distance functions, and learned SV seems like a reasonable/competitive baseline here?
> >    * it might be useful to report the K'  and FDE dimensionalities $d$s used for prefiltering the candidate sets (per your comment *"... we wanted to achieve end-to-end 0.84 recall, we would not use the raw output of the 20k dimensional FDE’s. Instead, we could use the (e.g.) 960 dimensional FDE"*), and how this affects final end-to-end recalls to help with understanding.
> >
> > * efficiency experiments: double checked the papers, it seems that we didn't use the same quantization scheme across ColBERTv2/PLAID vs this work. eg this work uses PQ-256-8 which compresses each float to 1 bit (line 307-308) whereas PLAID compresses each float to 2 bits ("For both vanilla ColBERTv2 and PLAID ColBERTv2, we compress all datasets to 2 bits per dimension, with the exception of MS MARCO v2 where we compress to 1 bit."). It would be helpful to control for related factors somehow, given retrieval efficiency is frequently memory-bandwidth bound.
> >
> > * Re Q5: thanks for pointing out fill_empty_clusters. But won't fill_empty_clusters break the FDE construction in Sec 2?

---

> > > ### Author Response · Authors · 2024-08-12
> > >
> > > We are glad that our explanation helped clarify some matters about our retrieval setup, and we will be sure to help clarify the exact setup further in the paper. We note that our main figure 1 (on page 2) shows the retrieval and reranking process, where the Chamfer reranking is a key step – we will add a further discussion on the importance of reranking in 1.2 where it is currently described.
> > >
> > > “multi-vector SV baselines”
> > > What does it mean to be a multi-vector SV baseline? We were slightly confused by this description and would like to understand better.
> > >
> > > “It might be useful to compare learned SVs which seem to work for small ds”
> > > We agree that training a SV model to approximate the Chamfer similarity is an interesting and important direction for future work. However, as discussed in the global response, the focus of our paper is to design an improved standalone retrieval algorithm for multi-vector databases. Like PLAID, we assume we are already given as input a multi-vector database (without necessarily having a training set needed for distillation), and we need to find the approximate nearest neighbors under the Chamfer Similarity for the MV embeddings in that dataset. Thus, comparing with other SV models, such as those distilled from a re-ranker, is somewhat out of the scope of the current paper.
> > >
> > > We also agree that pre-filtering with SV before a more complex or learned similarity is an popular and important method in the IR literature, and we will be sure to add references and discuss this alternative further in the paper.
> > >
> > > “multi-vector heuristics”
> > > We used the term “SV heuristic” to emphasize that the retrieval stage was only taking into account interactions between single vectors, and not aggregate interactions between sets of vectors (i.e. MV), which is what makes MV models unique. Moreover, we wanted to describe a *specific* heuristic which was to retrieve the top-k for each single vector and then aggregate them. We agree with the reviewer, though, that there may be a better term to describe this method, and will attempt to change it in the final version to something more descriptive.
> > >
> > > “it might be useful to report the K' and FDE dimensionalities “
> > > In the sentence on lines 332-336, we report that we used 10k-dimensional FDE’s with PQ-256-8 (32x compression) for all the latency experiments, and that we set K’ (=(# reranked)) to be equal to the beam width W (which is variable, since we tune so that our recall matches the recall of PLAID for each dataset). We will be happy to add the specific value of the beam-width K’ = W used for each of the six datasets in Figure 7 to give a more complete picture of the parameters.
> > >
> > > “quantization scheme across ColBERTv2/PLAID vs this work”
> > > We note that the standard PQ technique we use in this paper is different than the 1-2 bit compression scheme used in PLAID/ColBERTv2. Standard PQ-256-8 compresses chunks of size 8 into one of 256 centroids, whereas ColBERT/PLAID used many more centroids (2^32 or 2^16) plus a residual with 1 or 2 bits per dim. One advantage of PQ-256-8 is that using fewer centroids allows us to pre-compute a codebook that will fit in cache, allowing for very fast scoring (whereas 2^16 centroids would be too large of a codebook to fit in cache).
> > >
> > > “It would be helpful to control for related factors somehow”
> > > We note that we do conduct QPS experiments in C.4 considering different compression schemes, including PQ-256-4 which would compress blocks for 4 coordinates into one byte (so 2 bits per float). We found PQ-256-8 to offer a better QPS vs Recall tradeoff, since scoring is faster with a more compressed vector (and not much recall is lost). Since there are several differences between our approach and PLAID — (1) we are using a graph-based index for retrieval instead of IVF, which have different memory access patterns (IVF enjoys more sequential accesses) and (2) we are doing PQ mainly on the FDE’s and they are doing the PQ on the original 128-d vectors — it would be difficult to compare the two methods in precisely the same setting. However, our PQ-256-8 and PQ 256-4 with a similar number of retrieved candidates per query vector (and using 10k-dim vectors) would give the same number of bits retrieved from memory on MS Marco as PLAID (although note that our FDE vs. SV Heuristic section shows that FDE’s need to retrieve fewer candidates to get the same recall).
> > >
> > > “won't fill_empty_clusters break the FDE construction in Sec 2?”
> > > We would like to emphasize that fill_empty_clusters is part of Section 2. Thus, it is used in the theorems that we prove and in all of our experimental results. Thus, far from breaking the FDE construction in Sec 2, fill_empty_clusters is actually a core part of the FDE construction in Sec 2.
> > >
> > > We thank the reviewer again for this valuable discussion. If the reviewer agrees with the above points of clarification, we encourage them to reevaluate their score.

---

### Official Review · Reviewer_b2ZE · 2024-07-03

**Soundness:** 3
**Presentation:** 3
**Contribution:** 2
**Rating:** 4
**Confidence:** 3

**Summary:**

Efficient vector retrieval to maximise inner product similarity is well studied, and this paper explores the issue of multi-vector retrieval to support late interaction models like Colbert. The core idea is to use SimHash to generate clusters of multiple representation of documents, represent each document's vectors in a cluster using a centroid, and use that with the centroid of query vectors that land in the same cluster. Authors rely on the SimHash's approximation theory to show that this approach approximates the maxSim measure used by Colbert. Experimental results indicate Muvera outperforms PLAID in some of the datasets from BEIR in terms of query latency and Recall@k.

**Strengths:**

s1. Simple yet effective idea that seems to work on a variety of datasets.
s2. Well written, and experiments are well conducted.

**Weaknesses:**

w1. the theoretical analysis of the presented approach is rather limited. SimHash theory focuses on EMD minimisation while the focus here is on Chamfer similarity maximisation. So, it would be good detail the theory further.

w2. despite multiple stages listed for PLAID, the performance of MUVERA is not particularly strong -- in terms of latency, PLAID outperforms MUVERA on largest dataset (MS-MARCO). It leads one to question if the observed latency gains are primarily due to small(er) dataset sizes? Similarly in terms of recall@k, MUVERA outperforms PLAID only in a couple of datasets.

w3. there is no mention of how far from ideal (i.e, ColbertV2 performance) is the proposed model?

**Questions:**

q1. I would have liked to see the connection with the EMD approximation theory of SimHash with the Chamfer distance approximation presented here. In general, it seems to me --authors can clarify if I am mistaken-- the key innovation is the use of centroids in each bucket, instead of the standard O(n^2) similarity computation done even with SimHash in applications such as shingling. It would also be useful to show the sensitivity to the value of B = number of partitions.

q2. Improvements over PLAID are not significant in my opinion -- in terms of both latency and recall @ k. There are only few datasets (NQ-{100,1000}, HotPotQA-{100, 1000} the recall is better, and latency is worse in ms-marco (a commonly used Colbert benchmark). Further, it would be also worthwhile comparing with the baseline of ColbertV2 (without using any of these efficient indexing techniques).

**Limitations:**

Given that the work is aimed at improving the performance of an existing retrieval framework, I do not expect to see any specific negative societal impact from the work. Authors also have given similar remarks.

---

> ### Author Rebuttal · Authors · 2024-08-05
>
> We thank the reviewer for their detailed comments and suggestions. We reply to the main questions and concerns below, and attempt to clarify several points.
>
> > W1: “SimHash theory focuses on EMD minimisation while the focus here is on Chamfer similarity maximisation”.
>
> We would like to point out that this is not accurate. SimHash as a Locality Sensitive Hash (LSH) function dates back to the paper “Similarity Estimation Techniques from Rounding
> Algorithms” (Charikar, STOC ‘02), where it is used to estimate the cosine similarity between single vectors. In fact, to our knowledge SimHash has not been previously used for theoretical guarantees for EMD in any work. Instead, theoretical results for sketching EMD, such as the seminal “Algorithms for Dynamic Geometric Problems over Data Streams” (Indyk STOC ‘04) or “Earth Mover Distance over High-Dimensional Spaces” (Andoni, Indyk, Krauthgamer SODA ‘08) use L1 distance LSH’s, which are based on random hypergrid shifts, instead of SimHash. However, if the vectors are normalized (as they are for us) then dot product similarity and cosine similarity are the same. This is why we use SimHash for our theoretical (and practical) algorithm, because Chamfer similarity is an aggregate over the max dot product / cosine similarity of each individual vector.
>
> > “So, it would be good detail the theory further.”
>
> Could you please clarify what part of the theory you would like to see be explained further so that we can do so? Note that our theorem proves an eps-approximation of the Chamfer similarity with the FDE dot product, which is essentially the best result one could hope for in a reduction from Chamfer similarity to dot-product similarity. Our method for embedding a set of vectors into a single vector is novel, and has not been used in prior work.
>
> > Q1:
>
> Firstly, we emphasize that the theoretical bound in our paper (Theorem 2.1) applies to the Chamfer Similarity, which is exactly the similarity considered in the rest of the paper. We reiterate that SimHash itself is a Locality Sensitive Hash function for cosine similarity as described above, not for EMD. Thus, there is no direct connection between our theoretical results and known sketching techniques for EMD. The only similarity to sketching techniques for EMD, which is mentioned in the paper, is that sketches for EMD also take the approach of embedding a set of vectors into a single vector using LSH, but the approach for doing so is dramatically different due to the significant differences between EMD and Chamfer (e.g., Chamfer is asymmetric and has no matching constraint).
>
> > “-the key innovation is the use of centroids in each bucket, instead of the standard O(n^2) similarity computation done even with SimHash in applications such as shingling. “
>
> While the usage of centroids for the construction of the document-side FDE’s is important, it is not the key innovation in our paper. Specifically, the goal of our paper is not to avoid a O(n^2) similarity computation (where n is the size of the query set), but rather to provably embed Chamfer Similarity into dot-product similarity. SimHash is used as a method to turn a multi-vector similarity (where different vectors can match with each other) to a single vector similarity, which fixes the ordering in which the coordinates align. The key idea is to use SimHash to partition the space so that close points always end up in the same bucket, and therefore the same block of coordinates and can be matched. For the theory, the centroid used in the document FDE could have been equivalently replaced with any point from the document that landed in that partition and the theoretical bounds would still apply. Another key insight novel to this work is the “fill_empty_partition” method, which assigns the nearest document point to any empty cluster to ensure that increasing the number of buckets does not decrease quality.
>
>
> > W2
>
> The dataset size alone does not explain the performance of PLAID vs MUVERA on MS-MARCO, because note that MUVERA significantly outperforms PLAID on HotpotQA (5.2M documents) and NQ (2.7M documents) which are both on the same order of magnitude as MS-Marco (8.8M Documents). Instead, as discussed in the paper, note that PLAID was highly optimized for MS Marco, and was the culmination of multiple papers (ColBERT, ColBERTv2, PLAID), that successively optimized retrieval on the MS MARCO dataset. Our method, on the other hand, is not fine-tuned in any way for MS MARCO, and the same parameters achieve good results on all of our datasets without re-tuning.  With regards to recall@k, in our latency experiments we set our beam width so that our recall matches the recall of PLAID, so that we can primarily compare the latency at a given recall. The goal in these experiments was not to outperform PLAID’s recall, but rather to outperform it in latency at a fixed recall.
>
>
> > W3: “there is no mention of how far from ideal (i.e, ColbertV2 performance) is the proposed model?”
>
> We actually do compare to the ideal ColbertV2 Performance (i.e., the performance obtained by a brute-force search using Chamfer similarity) in the paper. Specifically, in Figure 3 we show the baselines for “Exact Chamfer@N” for various N, which is precisely the brute-force ColbertV2 performance. The main insight from this figure is that one can over-retrieve with FDE’s and then re-rank to obtain near-ideal or ideal performance (i.e. matching ColbertV2) by using an FDE with recall value (given by the dots) above the corresponding line in the plot. We do this in our end-to-end experiments, where we achieve, for instance, where we obtain 90.2 R@100 and 97.1 R@1k for MS MARCO (Figure 7), which can be compared to 91.4 and 98.3 for ColBERTv2 (as reported in Figure 3 and also in the PLAID paper). Interpretation of Figure 3 is further discussed in our response to reviewer Z5yi.

---

> > ### Comment · Reviewer_b2ZE · 2024-08-13
> > **Re: Rebuttal by authors**
> >
> > I thank the authors for providing detailed responses to my questions.
> >
> > > SimHash as a Locality Sensitive Hash (LSH) function dates back to the paper “Similarity Estimation Techniques from Rounding Algorithms” (Charikar, STOC ‘02), where it is used to estimate the cosine similarity between single vectors.
> >
> > In (Charikar, STOC'02) EMD approximation using LSH is given in Section 4. I disagree that the paper dealt only with cosine similarity between single vectors. If you believe that your analysis is significant improvement or different over the analysis given there, please clarify.
> >
> > Since some of my other queries are dependent on this aspect, I would like to hear from the authors before I update my rating.
> >
> > > We actually do compare to the ideal ColbertV2 Performance (i.e., the performance obtained by a brute-force search using Chamfer similarity) in the paper. Specifically, in Figure 3 we show the baselines for “Exact Chamfer@N” for various N, which is precisely the brute-force ColbertV2 performance.
> >
> > Thanks for pointing this out. The computational cost for over-retrieving (upto 10k for reaching ColbertV2) and reranking is not given. latency and QPS plots seem to be for retrieving 1000 (please clarify if I am incorrect), and how does this over-retrieval compare with the performance of ColbertV2?

---

> > > ### Author Response · Authors · 2024-08-13
> > >
> > > We thank the reviewer again for this discussion and valuable comments.
> > >
> > > >”In (Charikar, STOC'02) EMD approximation using LSH is given in Section 4. I disagree that the paper dealt only with cosine similarity between single vectors. If you believe that your analysis is significant improvement or different over the analysis given there, please clarify.”
> > >
> > > Note that we did not claim that the (Charikar, STOC'02) only considered cosine similarity between single vectors. Instead, we stated that *SimHash*, which is a particular LSH function, was only used in that paper to estimate cosine similarity. For completeness: SimHash is a random hash function h:R^d → {0,1} which maps a d-dimensional vector to a single bit {-1,1} by (1) drawing a random gaussian vector g ~ R^d and (2) outputting the sign bit of the dot product <x,g>, so SimHash(x) = Sign(<g,x>). The reviewer is correct that the Charikar paper gives algorithms for sketching EMD, but they are *not* based on SimHash.
> > >
> > > Let us attempt to clarify the differences in the historical approaches to sketching EMD vs. the approach in our paper. In the Charikar paper, as well as all other papers on sketching EMD, the main technique is called probabilistic tree embeddings (e.g. Bartal trees or FRT embedding, see e.g. Bartal, Yair. "Probabilistic approximation of metric spaces and its algorithmic applications."). This technique creates a randomized embedding F from the original metric ( R^d for us, but also works for any metric) into a tree metric T, such that the shortest path distances in the tree T approximates the original metric distances. For two sets A,B, if we want to estimate EMD(A,B), we first apply F to each point in A and B, to get two sets of vertices F(A),F(B) in a tree. One then tries to compute the EMD in the tree; the key point is that EMD in a tree can be embedded isometrically into the L1 metric (over single vectors) via a simple folklore embedding. This is all fundamentally different from our approach, which does not use tree embeddings at all.
> > >
> > > The above is the general recipe used for all algorithms for sketching EMD. The Charikar paper obtained a log(n) log log(n) approximation for any general metric. Indyk (STOC ‘04) improved this to O(log n) for the 2-dimensional grid (R^2), and (Andoni, Alexandr, Piotr Indyk, and Robert Krauthgamer. "Earth mover distance over high-dimensional spaces." SODA 2008) extended this to a log^2(n) for high-dimensional spaces R^d. These last two algorithms used an L1 distance LSH based on random Quadtree decompositions (which splits the space into random nested hypergrids), which is fundamentally different from SimHash.
> > >
> > > For a more detailed overview of the history and techniques used in sketching EMD, see the overview Section 1.1, of (Rajesh Jayaram, Erik Waingarten, and Tian Zhang. "Data-Dependent LSH for the Earth Mover’s Distance." STOC 2024, https://arxiv.org/pdf/2403.05041).
> > >
> > > Now for our results, note that we surprisingly are able to prove strong sketching results without the usage of probabilistic tree embeddings at all. Further, the reviewer is correct that we get a much better approximation: specifically, we get a (1+eps) approximation instead of a O(log n) or O(log^2(n)) approximation. This is possible due to our different techniques and the many differences between EMD and Chamfer (e.g. Chamfer is asymmetric, and does not satisfy triangle inequality). Some of the key differences are (1) we use an *asymmetric* LSH (encoding documents different from queries) based on SimHash (2) we do not use a tree embedding approach, but instead partition the space on a *single* granularity (unlike the nested partitions in tree embeddings) (3) we embed Chamfer into single vectors with the *dot product similarity* instead of the “L1 distance” – these two spaces behave very differently. The combination of these three different techniques, as well as the difference in the function we are trying to approximation (Chamfer vs. EMD) is how we get the improved approximation.
> > >
> > > > “ The computational cost for over-retrieving... is not given ... how does this over-retrieval compare with the performance of ColbertV2?”
> > >
> > > We emphasize that the experiments the reviewer is discussing in Figure 3 are part of our *offline experiments* that are meant to compare how over-retrieval with FDE’s compares to brute force (using the ColBERTv2 Model). Since this paper introduces a new sketching method (FDEs) we used this experiment to show how well FDEs approximate brute force max-sim (chamfer).
> > >
> > > We note that we do spend substantial time in the paper discussing online retrieval experiments which show the tradeoff between recall and QPS (Figures 6 and 7, and more in the supplementary materials). These experiments compare our online retrieval solution to PLAID, which is the fastest retrieval mechanism for the ColBERTv2 model to date. Note that PLAID is orders of magnitude faster than brute-forcing ColBERTv2.

---

### Official Review · Reviewer_uq8U · 2024-07-07

**Soundness:** 3
**Presentation:** 3
**Contribution:** 2
**Rating:** 3
**Confidence:** 4

**Summary:**

The paper proposes a method of speeding up text retrieval approximating ColBERT multi-vector ranker with very high dimensional single vector using projections.

Authors demonstrate that the proposed approach is better compared to a colbert-based single vector heuristic and is comparable to PLAID in terms of accuracy and speed, but is easier to tune.

**Strengths:**

* The paper is relatively easy to follow (though using  many acronyms like SV, MV, FDE makes it harder)

* The approach is more efficient at retrieval compared to the SV heuristic baseline

* Authors show that the method does not require much parameter tuning to achieve results comparable to PLAID.

**Weaknesses:**

1) The heuristic approach is reminiscent of [Morozov, Stanislav, and Artem Babenko https://arxiv.org/pdf/1908.06887 ] where a more general problem is stated with a more general solution (using the ranker as the metric during the graph search). This approach seems extremely relevant, but not discussed/compared in the paper.

2) Using a single vector embedding search to approximate a more complex interaction sounds like distillation of a ranker to a retrieval, which is a well-known baseline (e.g. googling gives https://arxiv.org/pdf/2210.11708 or https://arxiv.org/pdf/2403.20327 ). As both ranker and retrieval are trained, it is not clear why not directly train the retrieval model to directly optimize the ranking objective. There might scenarios where distilling is not feasible for some reason, but for a research paper I think it is important to show how much of the performance is lost.

**Questions:**

I'd like the authors to address both points of weaknesses.

**Limitations:**

The approach is limited to multi-vector ranking as the function.

---

> ### Author Rebuttal · Authors · 2024-08-05
>
> We thank the reviewer for their comments and suggestions about the paper. Below, we address both points of weakness mentioned by the reviewer.
>
> > W1:
>
> We thank the reviewer for the reference to this paper. Using a re-ranker or general similarity metric in the graph Beam Search is an interesting approach, which has not been considered by any of the prior extensive literature on multi-vector models and retrieval that we know of. One important point to note is that scoring in the graph search becomes significantly more expensive when the re-ranking similarity is used (Chamfer for us). In contrast, our method uses heavily quantized vectors in the search (compressed by 32x), so that scoring the dot product between two 5k-dimensional FDE’s can be done in roughly the same time as the dot product between two 5000/32 ~ 150-dimensional vectors of floats, which is significantly faster than computing Chamfer similarity (which requires a (32 x 128) x (128 x ~78) matrix product). Furthermore, our approach is based on provable theoretical bounds, whereas it is not clear that searching directly with Chamfer similarity would work in graph based approaches (due to the non-metric aspects of the similarity).
>
> > W2:
>
> As discussed in the global author response, we would like to emphasize that the focus of our paper is to design an improved standalone retrieval algorithm for multi-vector databases. This is exactly the same goal as the PLAID paper. Thus the reviewer’s concern would apply equally to the (highly-successful) PLAID paper, whose retrieval approach is purely based on the multi-vector representations.
>
> Therefore, like PLAID, we assume we are already given as input a multi-vector database, and we need to find the approximate nearest neighbors under the Chamfer Similarity for the MV embeddings in that dataset. Thus, training different MV models, or comparing with other SV models, such as those distilled from a re-ranker, are both out of the scope of the current paper. We remark that the comparison of MV models versus other methods of retrieval have been extensively evaluated in the prior literature on multi-vector retrieval, and the continued extensive research on multi-vector models and retrieval is testament to the power of multi-vector models.

---

> > ### Comment · Reviewer_uq8U · 2024-08-13
> > **re: rebuttal**
> >
> > I would like to that the authors for the response and for scope clarification. I'll increase the soundness score
> >
> > I think the paper scope on adapting the existing multi-vector solutions limits its interest from a broader community, so I am going to keep my rating.

---

> > > ### Author Response · Authors · 2024-08-13
> > >
> > > We thank the reviewer for their comment, but disagree with the assessment on limited interest on this domain from a broader community! Importantly, we emphasize the following
> > >
> > > (1) The demand for neural information retrieval is stronger than ever. If the number of startups that are forming to offer this service, or the number of cloud service providers that are offering a variety of services that use information retrieval is not convincing enough, one can look at the sheer number of research papers that are being written in this domain. Here is a list of some of the papers that are posted to arxiv since the beginning of 2024 specifically on multi-vector: https://arxiv.org/abs/2407.20750 https://arxiv.org/abs/2405.15028 https://arxiv.org/abs/2404.02805 https://arxiv.org/abs/2404.00684 https://arxiv.org/abs/2402.15059 https://arxiv.org/abs/2403.13291 https://arxiv.org/abs/2402.03216
> > >
> > > (2) While a lot of headroom remains on the table in neural information retrieval, there hasn't been a lot of progress in the single vector modeling recently and it is believed that single vector models can’t address the remaining major challenges and multi-vector approaches can potentially move the needle here. These challenges include: information retrieval for ambiguous, multi-intent, and not-a-right-answer queries; information retrieval for complex and nuanced queries that require multiple different pieces of information to be answered.
> > >
> > > (3) In GenAI, which is undoubtedly the hottest field of AI at the moment, neural information retrieval is critical and is going to remain important in the foreseeable future in the RAG type systems. In this context, similar to (2) above, single vectors fall short in terms of quality for complex and critical problems such as multiple needles in a haystack problem that is in the center of many generative tasks.
> > >
> > > (4) While multi-vector models have already shown to outperform single-vector models for information retrieval, the main reason their use-case has not become standard in all the above mentioned domains is their lack of efficiency compared to single vector models and retrieval systems.
> > >
> > > We believe the above provides substantial evidence for the broad appeal of algorithms for multi-vector models

---

### Official Review · Reviewer_7tYU · 2024-07-13

**Soundness:** 4
**Presentation:** 3
**Contribution:** 3
**Rating:** 6
**Confidence:** 4

**Summary:**

Multi-vector representation can greatly help retrieval systems work efficiently and accurately, transforming these sets of multiple vectors into a single vector representation that can still encapsulate the information from the multiple vectors, allowing efficient search using traditional single-vector search techniques while preserving the benefits of multi-vector search for rich document representation.

**Strengths:**

Novelty and generalizability: by introducing Fixed Dimensional Encodings (FDEs) in MUVERA, which compress multi-vector data into a single vector. This concept is innovative as it allows for applications of single-vector retrieval techniques, such as MIPS, to multi-vector problems. And I can foresee how this work is generalizable to other tasks and other datasets.

Comprehensive experiment: this experiment fully compared the state-of-the-art techniques with careful experiment settings, as well as good methodology.

**Weaknesses:**

Computational overhead: as introduced by the FDE creation and query processing, the computational overhead may be high as the dataset grows. The MUVERA may have problems in preprocessing and dynamic updates.

The robustness of MUVERA across different types of retrieval tasks and its effectiveness in different domains (e.g., legal documents versus scientific articles) have not been extensively validated, for those domain-specific tasks, it could be beneficial to run experiments on different datasets from diverse domains.

**Questions:**

What are the main trade-offs in this work? Maybe dimension reduction and accuracy are one of them: it is advantageous for processing speed and memory usage, but it can also lead to information loss.

What are the specific challenges encountered when implementing FDEs in real-world systems, particularly concerning maintaining the balance between efficiency and retrieval accuracy?

Tuning parameter W in this work: "The only tuning knob in our system is W; increasing W increases the number of candidates
298 retrieved by MUVERA, which improves the recall" Can this be pre-determined or you can use a specific algorithm to decide it?

**Limitations:**

If the computation overhead is a barrier problem for MUVERA running on a very large dataset, it may face limitations on scalability of the FDE approach, especially in terms of how well it can be adapted to very large datasets or highly dynamic environments.

---

> ### Author Rebuttal · Authors · 2024-08-05
>
> We thank the reviewer for their comments and suggestions. We respond now to the specific questions of the reviewer.
>
> > “Computational overhead: as introduced by the FDE creation and query processing, the computational overhead may be high as the dataset grows. The MUVERA may have problems in preprocessing and dynamic updates.”
>
> We note that the cost of FDE generation is only incurred at index time, which can be done offline. Furthermore, the cost of generating an FDE from the multivector representation is linear in the size of the multivector representation, where the complexity scales linearly with the number of repetitions performed. We note that our latency measurements also include the cost of generating the FDE from the query’s multivector representation, and that the time spent on this step is less than 1% of the overall query time. Also note, with regards to dynamic updates, that the construction of FDE’s are data-oblivious, and therefore are easy to generate and maintain in a dynamic setting.
>
>
> > “Robustness of MUVERA across different types of retrieval tasks and its effectiveness in different domains (e.g., legal documents versus scientific articles)”
>
> We note that the BEIR retrieval benchmarks that we evaluate on, which are standard in the IR literature, are diverse, and include a mix of scientific documents, web queries, quora QA, and other types of domains. In this paper, we chose to focus on the same (diverse) set of datasets used in prior work on multivector representations.
>
> > “What are the main trade-offs in this work? Maybe dimension reduction and accuracy are one of them: it is advantageous for processing speed and memory usage, but it can also lead to information loss.”
>
> We agree with the reviewer that the relationship between FDE dimension and accuracy is the main trade-off that we optimize in this work. We further leverage compression techniques like product quantization to enable using higher-dimensional FDEs while keeping the index size for the FDEs moderately sized. Another trade-off that we explored is the relationship between the number of query embeddings and the search latency (via ball-carving) —to the best of our knowledge our work is the first to identify this relationship and we believe it may be of independent interest and importance in future work on multivector systems.
>
> > “What are the specific challenges encountered when implementing FDEs in real-world systems, particularly concerning maintaining the balance between efficiency and retrieval accuracy?”
>
> The main challenge is to tune the dimensionality of the FDEs to achieve a desired level of accuracy while bounding the space usage of the index. Thanks to the strong theoretical properties of FDEs, our implementation of FDEs is robust to specific choices in the FDE design space such as the type of partitioning used (as illustrated in Figure 3). After fixing a dimension that provides suitable accuracy, the next major challenge is to tune the quantization level to keep the index size manageable—as we show in the paper FDEs are remarkably robust to quite aggressive quantization, which enables the index size of MUVERA to be comparable to highly-engineered systems such as PLAID.
>
> > “Tuning W”
>
> W is a parameter that the user needs to tune to achieve a suitable level of recall—we note that this is standard practice in IR systems that are based on graph-based approximate nearest neighbor search. Understanding how to automatically set W to achieve a given level of recall seems to be equivalent to obtaining provable guarantees for these ANNS methods which is a major open question.

---

> > ### Comment · Reviewer_7tYU · 2024-08-12
> > **Some additional questions**
> >
> > Thanks for the clarification.
> >
> > I have some additional questions for this work:
> >
> > - first, in terms of FDE generation, due to the fact that I don't have your code access, I'm curious how long it takes to finish FDE generation on your datasets provided.
> >
> > - Second, the data diversity I mentioned may have another concern, which is skewed data distribution, can this framework somehow overcome this problem theoretically?
> >
> > - how big is the index in terms of the disk usage on some specific dataset? Do you need to load the index into memory like HNSW indexes? Your index disk usage and memory footprint may be a crucial point of your work.
> >
> > - On the other hand, building an index on a multi-vector is somehow a fixed solution for specific multi-vector queries, e.g., we have vector columns vc1, vc2, vc3, vc4, and we somehow build them together via FDE, your framework is focused on the combinational query on vc1, vc2, vc3, vc4, right? Can it be used to serve queries on vc1 + vc3?
> >
> > - Does the W parameter have a relatively monotonic effect on the results? For example, in IVF indexes, the larger the search partition parameter is set (n_probe), the higher the recall rate will be, and the longer the latency will also be expected.

---

> > > ### Author Response · Authors · 2024-08-13
> > >
> > > Thanks for your further comments and questions.
> > >
> > > > “first, in terms of FDE generation, due to the fact that I don't have your code access, I'm curious how long it takes to finish FDE generation on your datasets provided.”
> > >
> > > FDE generation is extremely fast—the average running time to generate an FDE is roughly 0.001s, and is embarrassingly parallelizable.
> > >
> > > > “skewed data distribution, can this framework somehow overcome this problem theoretically?”
> > >
> > > With regards to skewed data distributions, we would like to emphasize that our theoretical results hold for any input, even worst case inputs, as is standard in theoretical computer science. Thus skewed data would not affect the performance of our algorithm theoretically.
> > >
> > > > “how big is the index in terms of the disk usage on some specific dataset? Do you need to load the index into memory like HNSW indexes? Your index disk usage and memory footprint may be a crucial point of your work.”
> > >
> > > We provided some details about the index size in our discussion with reviewer Z5yi. The index size for MUVERA is dominated by the space used for the multivector embeddings. As we discussed with reviewer Z5yi, future optimizations for our system could reduce the space usage by applying quantization techniques to the multivector embeddings. We note that the space used for our graph index and FDEs is about 10% of the space of the original multi-vector representations. We would be happy to add exact space usage numbers for the index to the final version of the paper.
> > >
> > > Yes, we do load the index into memory as in an HNSW index. We are using the DiskANN algorithm fully in-memory, which we will emphasize in the paper. The DiskANN algorithm is extremely competitive and many vector search companies (e.g., Pinecone) have migrated from HNSW to DiskANN due to its strong performance.
> > >
> > > > “building an index on a multi-vector is somehow a fixed solution for specific multi-vector queries, e.g., we have vector columns vc1, vc2, vc3, vc4, and we somehow build them together via FDE, your framework is focused on the combinational query on vc1, vc2, vc3, vc4, right? Can it be used to serve queries on vc1 + vc3?”
> > >
> > > We are not fully sure what the reviewer means by this question—if we understand the reviewer’s question correctly, the reviewer is describing a vector database use case where an object has multiple vector columns (v1, .., v4), where the vectors are produced using potentially different single-vector models, which is fundamentally different from the setting that we consider.
> > >
> > > Our paper is focused on the multivector setting where we have a single multivector *model* which transforms an object into a set of vectors, which all live in the same latent space. When a query arrives, we run the same model on it to transform it into a set of vectors. The goal is to find the most similar document, where similarity is given by a specific set-set function—the chamfer similarity.
> > >
> > > > “Does the W parameter have a relatively monotonic effect on the results? For example, in IVF indexes, the larger the search partition parameter is set (n_probe), the higher the recall rate will be, and the longer the latency will also be expected.
> > >
> > > Yes, increasing W monotonically improves the quality of the results—it will result in higher recall at the expense of higher latency as the reviewer correctly pointed out.

---

### Official Review · Reviewer_Egyf · 2024-07-13

**Soundness:** 2
**Presentation:** 2
**Contribution:** 2
**Rating:** 3
**Confidence:** 3

**Summary:**

The paper presents a method of converting multi-vector query-document retrieval problem to a single query-document vector-based retrieval problem. The basic idea is to project the multi-vector queries and documents into fixed-dimensional embedding through random projections. Further efficiency in storage of the random projections is achieved through vector quantization (product quantization) methods. The method is shown to result in improved recall as well as improvements in time performance. The projection into the fixed-dimensional space is obtained by latent semantic hashing (LSH). This concept could have been explained in a much simpler manner than presented in the paper.

**Strengths:**

The formulation of the problem and the theoretical proofs are the strength of the work. The experimentation has been done on the obligatory BEIR benchmark datasets. Comparison has been made to one other approach (PLAID).

**Weaknesses:**

It seems to me in the current age of LLMs, this technique appears to be a bit dated. There wasn't any mention of SpladeV2 and its variants either that are actively being used in commercial vector databases along with vector quantization in billion-vector size databases. Addressing this would be good for related work.  A 0.6 sec time performance for a single search is too slow for commercial uses, since typically multiple searches are rolled into an overall one.

It is difficult to reproduce this work from the level of detail provided. A block diagram illustrating the overall steps would be useful.

There are typos in the draft so a good spell check is needed.

**Questions:**

The fact that LSH preserves nearness is known. How does adding product quantization alter this ability? Does this technique allow for vector search using the compressed representation or the vectors need to be decompressed back before similarity search?

**Limitations:**

The limitations of the algorithm have not been adequately addressed. For example, for what sized datasets is this technique suitable or will be an overkill?

---

> ### Author Rebuttal · Authors · 2024-08-05
>
> We thank the reviewer for their comments and suggestions. We respond now to the specific questions of the reviewer.
>
> > “It seems to me in the current age of LLMs, this technique appears to be a bit dated. There wasn't any mention of SpladeV2 and its variants either that are actively being used in commercial vector databases along with vector quantization in billion-vector size databases.”
>
> We would first like to emphasize that multi-vector models are a fairly new technique which have attracted considerable attention in the last few years, especially in the last 2 years, with many papers being published on the topic at each major conference (see the large list of references in the paper for a subset of these works).  Thus, it does not seem to be the case that multi-vector models are outdated in the current LLM era.
>
> Furthermore, while LLMs have shown amazing results in generative tasks, they are not really strong for information retrieval tasks out of the box and, for example, using standard techniques such as special token pooling or average pooling on top of LLMs for information retrieval from large corpora of text doesn’t match up with even older and simpler semantic encoder approaches. As a result, a contrastive loss based tuning is applied to embeddings pooled from LLMs to make them useful for information retrieval (e.g. Sentence T5, Gecko). Multi-Vector training over LLMs is an even more recent and active field of research that has been challenging researchers in academia and industry both on the modeling and on the efficiency worlds (e.g. XTR, ALIGNER, ColBERT, ColBERTer, ColBERTv2).
>
> Finally, we emphasize that the goal of our work (like PLAID) is to design a more practically- and theoretically-efficient method for multi-vector retrieval. We do not focus on modeling aspects, or comparing multi-vector with other retrieval methods such as SPLADE or other single-vector methods. Such comparisons have already been considered extensively in the literature.
>
> > “A 0.6 sec time performance for a single search is too slow for commercial uses, since typically multiple searches are rolled into an overall one.”
>
> We would like to emphasize that these latency experiments were run on a single core, as is standard practice in the line of IR papers our work contributes to (ColBERT, ColBERTv2, PLAID). In commercial settings, queries are typically run using more cores, and the queries are distributed across multiple machines resulting in significantly faster searches. We focused on single core evaluation as this is the standard methodology for measuring latency in the information retrieval literature (e.g., PLAID and other IR papers all use single core measurements when reporting latency).
>
> > “A block diagram illustrating the overall steps would be useful.”
>
> We include block diagrams of the overall steps of our algorithm in Figure 1; if the reviewer has additional suggestions on how to improve our figure we would be happy to incorporate them.
>
>
> > “The fact that LSH preserves nearness is known. How does adding product quantization alter this ability?”
>
> Firstly, we would like to point out that the product quantization is being done on top of the FDEs, not on the original vectors before they are added to the FDEs. Since LSH is only used to decide how to add initial vectors to the FDEs, product quantization does not interact with the LSH at all in this way. Nevertheless, for completeness we remark that even if we did do PQ first before applying LSH, the theoretical and practical guarantees of LSH would still hold. This is because PQ approximates the original vector by a very nearby vector (which is represented in compressed form), therefore preserving nearness to the original vector. So even if PQ was applied before LSH, the guarantees of LSH would still apply.
>
>
> > “Does this technique allow for vector search using the compressed representation or the vectors need to be decompressed back before similarity search?”
>
> Yes, the vector search and scoring is done using the compressed representations. This is one of the main benefits of product quantization – you can score similarity even faster than a fixed dot product using asymmetric scoring – this is a standard technique in similarity search.

---

> > ### Comment · Reviewer_Egyf · 2024-08-13
> > **Response to rebuttal**
> >
> > Thanks for those clarifications. So could you comment on how your method compares to Splade V2?
> > Also, what would be the time performance in realistic settings? What is the largest vector database you have tested this against?
> > Finally, it would be good to compare the results of retrieval with and without PQ and LSH components to see how much loss in accuracy.
> >
> > So although the response addressed my questions somewhat, the answers I am looking for are still not there as indicated above.

---

### Official Review · Reviewer_4i1U · 2024-07-14

**Soundness:** 3
**Presentation:** 3
**Contribution:** 3
**Rating:** 6
**Confidence:** 3

**Summary:**

This paper aims to improve the search efficiency of multi-vector retrieval models, such as ColBERT. Specifically, the authors propose MUVERA framework, which reduces the multi-vector (MV) similarity of a query/document pair to the single-vector (SV) similarity by constructing Fixed Dimensional Encoding (FDE) of MV representation. Such reduction allows re-using many highly-optimized MIPS solvers for multi-vector retrieval. The authors also provide theoretical guarantee on the approximation error of FDEs. Empirically, compared to previous SOTA heuristic for multi-vector retrieval, MUBERA can achieve a similar Recall while significantly reduce the latency.

**Strengths:**

- The proposal of Fixed Dimensional Encodings (FDEs) such that its inner product approximate the multi-vector (MV) similarity is novel and interesting
- The authors also provide strong theoretical guarantee under $\epsilon$-approximation to the true MV similarity
- Promising empirical results that significantly reduce the latency on most BEIR datasets compared to PLAID

**Weaknesses:**

Here are the summarized version. See detailed comments/questions in the next Section.
- The index size of MUVERA framework can be quite large
- MUVERA has sub-optimal performance on the MS-Marco dataset
- no released code

**Questions:**

### Q1: The index size of MUVERA
As illustrated in Figure 1, MUVERA consider a two-stage search procedure: (1) first using MIPS to obtain a candidate set of documents; and (2) re-rank the candidates with exact Chamfer similarity.
- (a) For the first step, using PQ codes on the FDE of documents and build ANN index indeed save the memory. For the second step, however, computing the exact Chamfer similarity still require storing all the original token embeddings per document?
- (b) What's the memory usage and index size for stage 1 and stage 2, respectively?

### Q2: Why do MUVERA has sub-optimal performance, compared to PLAID, on the MS-Marco dataset?
- (a) Is it because of using data-agnostic partitioning functions (SimHash), making the approximation error of FDE larger?
- (b) If the token embedding distributions are skew and concentrating on certain sub-space (maybe this is the case in MS-Marco?) , is it still a good idea to use SimHash in FDEs? Or k-means would do a better work?

### Other minor comments
- (a) In Figure 1 caption, there is a typo: "comapred" => "compared"
- (b) At Line 133: "vectors that are closer are more are more likely to..." => there seems to be some repeatedly wording
- (c) In Figure 2, the legend on the left-hand-side seems wrong? The orange square should be Doc Embeddings and the blue circle should be Query Embedding, according to the right-hand-side plot?

**Limitations:**

To my knowledge, there's no potential negative societal impact.

---

> ### Author Rebuttal · Authors · 2024-08-05
>
> We thank the reviewer for their comments and suggestions. We respond now to the specific questions of the reviewer.
>
> > Q1(a):
>
> We first clarify questions about the index size of MUVERA. The reviewer is correct that we need to store both (some representation of) the FDE’s and original multi-vector (MV) representations, however we do not need to fully store either of them. Specifically, we can use the same product quantization for the original MV representations that we use for the FDE’s. Since the focus of the paper was on latency (and not necessarily space) and the new FDE method, we did not run our end-to-end evaluations with the compressed MV representations when rescoring. However, doing so would (1) speed up rescoring (since asymmetric PQ scoring is up to an order of magnitude faster than scoring the uncompressed vectors) and (2) result in negligible quality loss. For concrete numbers on (2), we found that using PQ256-2 (i.e. 8x compression) on the original 128-d vectors resulting in *no downstream quality loss* in the recall for Chamfer scoring (i.e. uncompressed R@100 = 91.49 vs compressed = 91.63, or 98.44 uncompressed vs 98.47 compressed for R@1k). If we compress using PQ256-4 (16x compression), we see very negligible quality loss (the recall@N is 91.1 and 98.38 for N=100,1000 respectively). Also note that PLAID and ColBERTv2 also store compressed versions of the original MV embeddings (via centroids + small residual), which could easily be used for rescoring in our case as well.
>
> > Q1(b):
>
> for concrete numbers, when using 10k dimensional FDE’s on MS MARCO (which originally has an average of 78.8*128 = 10086 floats per MV representation), we used PQ256-8 (as reported in the paper) to compress the vectors by 32x, and the full index size is 11 GB (instead of 340GB to store the uncompressed FDE’s). For the original multi-vector representations, the uncompressed index size is about 340 GB on disk. Using PQ256-4 we reduce it to 42GB, and using PQ256-8 it goes down to 21 GB (with negligible quality loss as described above). Thus the total index size for stages (1)+(2) would be 11 + 21 = 32 GB for MS Marco, which is more than a 10x compression from the original vectors, and comparable to PLAID’s index (21.6GB).
>
> > Q2:
>
> We first want to emphasize (1) that we outperform (sometimes significantly) PLAID on all 5 other datasets we studied, in terms of both latency (and sometimes recall too), and (2) PLAID only marginally outperforms our method on MS MARCO. As discussed in the paper, PLAID was highly optimized for MS MARCO due to the prevalence of that dataset, and its parameters were likely overfit. Quoting from a recent reproducibility study of PLAID (https://arxiv.org/pdf/2404.14989): “PLAID’s Pareto frontier is a patchwork of parameter settings; changing one parameter without corresponding changes to the others can result in slower retrieval with no change to effectiveness.” In contrast, using one single set of parameters for all datasets, we get strong results across the board with MUVERA. We also want to emphasize that PLAID was the third in a series of papers (ColBERT, ColBERTv2, PLAID) which attempted to highly optimize the SV Heuristic. MUVERA, on the other hand, is the first paper on the FDE-based approach for MV retrieval, and thus has not seen the same level of optimization. We believe the fact that we match (and often exceed) PLAID on many datasets with this new approach without heavy systems optimization is a strong benefit of our work.
>
> > Q2(a) + (b):
>
> With regards to Simhash vs. K-means: As shown in Figure 3 (the grid-search), SimHash outperforms k-means on MS MARCO for every parameter range on the pareto frontier. The reason for this is as follows: because we need the FDE’s to be small dimensional, we cannot set the number of partitions B to be too large, so we often use B = {16,32,64}. For such small values of B, you cannot hope to capture the global behavior of a dataset of 8.8M * 77.8 ~ 684M vectors in MS MARCO. Thus, using only <100 centers from k-means on nearly a billion vectors will give very little information about any individual datapoint’s MV representation (note we cannot run a different k-means on each document’s MV representation, since the partition needs to be the same for all points!). Thus, data dependent techniques will not help much here. Instead, SimHash likely performs better because it provably is an LSH and gives good approximate partitions (unlike k-means, which has no theoretical guarantees) for *all* points in the dataset. With regards to the question about skew in the dataset – the data would have to be incredibly skewed (i.e. tightly concentrated around 100 clusters) for k-means to work well. Additionally, each MV representation would have to be spread out over the different clusters (otherwise the partition would not split it into smaller parts).
>
> > “(c) In Figure 2, the legend…’
>
> This is a very good catch! We thank the reviewer for finding this mistake in the diagram, as well as the other typos mentioned. We will fix them right away.

---

> > ### Comment · Reviewer_4i1U · 2024-08-11
> >
> > Thanks for the clarification about the index size. I will encourage the author also supplement the index size of MUVERA (stage1 and 2 respectively) together with the Recall and latency results, in the main experiment sections. The index size is also a crucial factor to be considered when deploying the ANN search system for online retrieval. Thus, I will like to keep my score as is.

---

### Author Rebuttal · Authors · 2024-08-05

We thank all the reviewers for their thoughtful comments and suggestions. We reply to the questions and comments of each reviewer individually in the corresponding rebuttal fields. First, we would like to emphasize here several global points which will address common concerns of the reviewers.

Firstly, we stress that the goal of our work is to design a more practically- and theoretically-efficient method for multi-vector retrieval. We do not focus on modeling aspects, or comparing multi-vector with other retrieval methods such as SPLADE or other single-vector methods. Such comparisons have already been considered extensively in the literature. Like PLAID, our method is a standalone multi-vector retriever, is independent of the underlying multi-vector model, and works given any database of multi-vector embeddings. Therefore, while there may be other approaches to IR not based solely on the multi-vector representations, they are out of the scope of this paper.

Secondly, we would like to make clarifying points with respect to the comparison of MUVERA and PLAID. While our goal is to design a state of the art retrieval system for multi-vector search, we see the main contribution of our work as contributing a new approach to multi-vector search which is based on principled theoretical guarantees, as an alternative to the prior heuristic approaches used by other work on multi-vector retrieval. PLAID itself is a highly-engineered system which is built on a sequence of successive optimizations (ColBERT, ColBERTv2, PLAID), and has been carefully hyper-optimized, especially for MS MARCO. In fact, quoting from a recent reproducibility study of PLAID (https://arxiv.org/pdf/2404.14989): “PLAID’s Pareto frontier is a patchwork of parameter settings; changing one parameter without corresponding changes to the others can result in slower retrieval with no change to effectiveness.”

In contrast, our paper is the very first of its kind using the new FDE method, and is significantly simpler to tune and, as a result, is more straightforward to generalize to arbitrary models and data sets without significant quality regression. We used the *same parameter settings* for all six of the datasets tested, and on five out of the six we outperformed PLAID. On MS MARCO, which PLAID was highly tuned for, we nearly matched its performance, which we find to be a strong contribution given that our technique has not benefited from the same extensive engineering as PLAID.

Finally, we emphasize that in our offline experiments, we show that as a method for retrieval FDE’s are 2x-5x more efficient than the SV heuristic (Figure 5), meaning that they need to retrieve many fewer candidates to achieve the same recall. Since PLAID uses (an optimized version of) the SV heuristic, we believe that this signals that there is significant headroom for optimizing the approach of using FDE’s for multi-vector retrieval.

---

### Comment · Area_Chair_a4qx · 2024-08-12
**Reviewer-Author Discussions**

Dear reviewers: as you are aware, reviewer-author discussions phase is ending on Aug 13. We request you to kindly make use of the remaining time to contribute productively to these discussions. If you have not read and/or responded to author rebuttal, please do it asap so that the authors get a chance to respond to you. If you have more questions to ask or want further clarification from the authors, please feel free to do it.

Dear authors: please read the responses of the reviewers and if there are any pending questions/request-for-clarification, please attend to them.

---

### Decision · Program_Chairs · 2024-09-25

**Decision:**

Accept (poster)

**Comment:**

This work addresses the problem of improving the efficiency of multi-vector similarity search in the context of multi-vector models like ColBERT that represent queries as well as documents using multiple embedding vectors. Previous work addressed the efficiency problem of multi-vector similarity search taking a heuristic approach without any theoretical guarantees. In contrast, the current work proposes an approach that is theoretically grounded, simple to implement and use, relatively less sensitive to parameter setting and gives competitive retrieval performance and latency compared to the heuristic approach.

The main idea here is to transform the multi-vector similarity search problem into a single-vector similarity search problem using Fixed Dimensional Encodings (FDEs) of queries and documents. FDEs are designed so that their inner product closely approximates multi-vector similarity as measured using Chamfer similarity and thereby enabling fast multi-vector retrieval using off-the-shelf MIPS solvers. Further, data oblivious transformations are used to compute FDEs thereby eliminating the need for additional datasets and/or training for the transformations. As FDEs are high-dimensional and therefore can have a large memory footprint, the work employs a vector compression technique called product quantization and thereby compressing the FDEs significantly without incurring substantial loss in recall.

The work provides theoretical guarantees for the proposed approach. It shows that the inner product of FDEs of two multi-vector sets is an ε-approximation to the Chamfer similarity of the two sets.

The work compares the end-to-end retrieval performance of the proposed approach, as measured using Recall@N, with the heuristic search algorithm PLAID on MS-MARCO and several other datasets. While it achieves 10% higher recall an and  90% lower latency on average compared with PLAID, it has significantly higher latency (0.31 s vs 0.221 s) and nearly the same recall (0.971 vs 0.975) on MS-MARCO.

The problem and the proposed approach are well-motivated. The solution proposed is principled and backed by theoretical guarantees. Memory footprint is reduced significantly by a quantization trick. Results are very promising and competitive. The paper is well written and has sufficient detail to reproduce the experimental results. Authors have promised to release the code. Overall, the work makes significant contribution to multi-vector similarity search.

The experimental study can be strengthened further by addressing the following issues:

	1. Latency comparison is made only for 1 CPU thread. Speedup on GPU should also be reported.
	2. End-to-end evaluation reports only Recall@N. MRR should also be reported as done by PLAID [35].
	3. End-to-end evaluation is not done on LoTTE benchmark.
	4. Investigation of the reasons for higher latency for MS-MARCO is missing. A study of latency as a function of dataset size could be useful in this regard.